**communications** sustainability

# Technology-driven reduction of fish post-harvest loss could enhance food security and economic resilience
Haizhou Wu [1,2] ✉, Jingnan Zhang [1], Heng Zhu [3], Omar Peñarubia [4] & David F. Willer [5] ✉

Globally, only 54% of harvested fish is consumed directly by people, with the remainder lost to spoilage, inefficient processing, limited by-product utilization, or diverted to non-food uses. This inefficiency limits the nutritional, economic, and environmental potential of aquatic foods. Here, we assess the impact of targeted post-harvest interventions—including cold chain improvements, better handling practices, and valorisation of by-products, using a quantitative modelling approach with a qualitative synthesis of case studies and literature. We show that increasing net fish consumption by humans to 74% through feasible technological adoption could deliver an additional 850 million portions of fish per day, without harvesting a single extra fish. These "hidden harvests" could meet global dietary protein and micronutrient needs while reducing price to the consumer by nearly 10%. Whilst these findings should be seen as upper limits rather than expected outcomes. They highlight post-harvest optimisation as a critically underutilised lever for advancing nutrition security, reducing pressure on aquatic ecosystems, and achieving sustainable, equitable growth in blue food systems. Reducing waste, not simply increasing catch, is the key.

## Main

Fish and seafood are critical components of global food and nutrition security[1], yet substantial post-harvest losses (PHL) significantly undermine their potential benefits. Approximately 35% of the global fish catch is lost or wasted along supply chains, translating into roughly 10–12 million tons annually[1,2]. These losses, driven largely by spoilage, inadequate cold storage, and inefficient handling and transport, impose economic costs while limiting opportunities to strengthen food security, improve fisherfolk livelihoods, and conserve fishery resources[3,4]. Technological innovations—from simple handling improvements to advanced processing—offer promise for reducing PHL[1,5].

The urgency of addressing PHL is heightened by rapid growth in global fish consumption (a 127% increase over the past three decades), driven by demand for protein-rich, micronutrient-dense foods[1,6]. Seafood provides over 3.2 billion people with at least 20% of their animal protein intake and supplies essential nutrients such as omega-3 fatty acids, vitamins, and minerals[3,7]. However, with more than one-third of wild fish stocks exploited beyond sustainable levels, and aquaculture facing its own environmental and resource constraints, increasing production alone cannot meet rising nutritional needs—particularly as climate change and population growth intensify pressures on aquatic food systems[1,3].

Historically, PHL have received less attention than production-focused improvements in fisheries and aquaculture[1,8]. Research and policy have prioritised increasing catch volumes or farm outputs while largely neglecting substantial PHL[1,2]. Similarly, losses in aquaculture remain understudied despite comparable inefficiencies, leading to critical data gaps that hinder targeted interventions and comprehensive policy development[8,9]. Technological innovations offer solutions to address these often-overlooked losses across the entire value chain, from catch to consumption. Solutions range from simple, low-cost preservation methods in low-income tropical fisheries to advanced refrigeration and packaging technologies in the industrialised fisheries of high-income regions[9–11]. While no single approach is universally applicable, all aim to increase the proportion of fish retained within the food system and ease pressure on fish stocks.

¹Hubei Technology Innovation Center for Meat Processing, College of Food Science and Technology, Huazhong Agricultural University, Wuhan, Hubei, China. ²College of Animal Science & Technology, College of Veterinary Medicine, Huazhong Agricultural University, Wuhan, Hubei, China. ³Department of Mechanics and Maritime Sciences, Division of Marine Technology, Chalmers University of Technology, Gothenburg, Sweden. ⁴Fisheries and Aquaculture Division, Food and Agriculture Organization of the United Nations (FAO), Rome, Italy. ⁵Department of Zoology, University of Cambridge, Downing Street, Cambridge, UK. ✉e-mail: haizhou@mail.hzau.edu.cn; dw460@cam.ac.uk

Despite promising developments, knowledge gaps remain about how much additional fish supply can be recovered and the impacts on nutrition, economics, and conservation. To address these gaps, our study integrates quantitative modelling with qualitative case studies to evaluate the potential for recovering fish supply through technological interventions, assess implications for nutrition and public health, and analyse economic benefits. Region-specific examples—from artisanal fisheries to industrial operations—highlight diverse challenges and opportunities, underscoring the global relevance of reducing PHL. By integrating these insights, we aim to inform targeted strategies, support policy and investment, and demonstrate that reducing PHL can deliver wide-ranging benefits for both people and the planet.

## Results

The following results quantify the system-level potential of post-harvest technological interventions to increase the availability, nutritional value, and economic efficiency of aquatic foods. We first examine how technological adoption alters the fraction of harvested fish ultimately utilised for direct human consumption. We then assess the nutritional implications of recovered biomass using a representative, nutrient-rich species as an illustrative upper envelope. Finally, we explore the economic consequences of improved utilisation through a simplified cost-structure framework. It should be noted that rather than predicting realised outcomes, the analysis is designed to establish upper bounds on what could be achieved under idealised and harmonised adoption conditions, thereby bounding the scale of opportunity embedded in post-harvest optimisation.

## Extent of fish post-harvest losses from global and regional perspectives

Our model quantifying the transformation from harvest to consumption is shown in Fig. 1, and here we highlight the three key stages: allocation between food and non-food applications, post-harvest loss and waste, and by-product management[4,7]. On a global level today (baseline, 0% technology adoption, Fig. 2b) 54% of harvested fish is directly consumed by people (including consumption of reutilised by-products). For the remaining 46%, 11% is direct non-food use (e.g., fishmeal, fish oil, pet food), 18% is by-products from direct human consumption that are not reutilised for direct human consumption, and 17% is direct loss and waste earlier in the value chain[1,4]. This total wastage of 17% plus an additional 29% not being used directly for human food is in line with the FAO's global food loss estimates for fisheries, which range from 25% to 35%[4,12]. It is important to note that not all unutilized fish are classified as food loss. Non-food utilisation streams (e.g., reduction to fishmeal and fish oil) remain productive uses and are therefore distinct from post-harvest loss, which refers only to edible biomass that fails to reach consumers.

However, significant regional variation exists due to differences in processing infrastructure, cold chain availability, and market access[1,3,4]. Figure 3 highlights these disparities: while developed regions generally achieve high overall utilisation rates, many developing regions, particularly in the Global South, experience substantial inefficiencies along the supply chain. These findings point to a clear need for targeted, context-specific interventions to improve the conversion of harvested fish into consumable food. With these interventions (discussed below), and thus the resultant increases in technology adoption levels, the industry could deliver the increases in direct human consumption shown in Fig. 2c–f, up

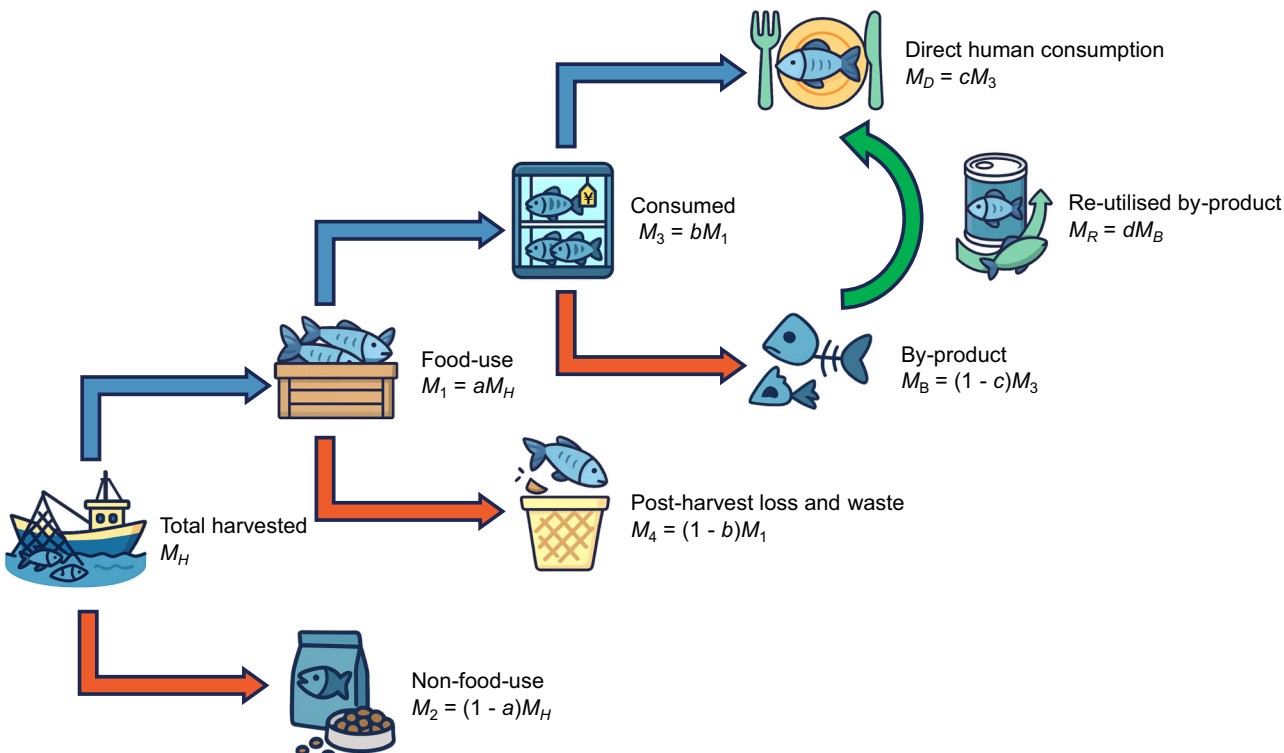

**Fig. 1 | Conceptual framework of fish utilisation in the post-harvest value chain.** This diagram illustrates the allocation of total harvested fish biomass ($M_H$) across food and non-food uses, as well as subsequent pathways within the food-use chain. A share $a$ of the total harvest is allocated to food-use ($M_1 = aM_H$), while the remainder ($M_2 = (1-a)M_H$) is diverted to non-food uses such as fishmeal and fish oil. Within the food-use stream, a fraction $b$ of biomass is retained as edible product after processing and reaches the consumption stage ($M_3 = bM_2$), while the remaining $(1-b)M_2 = M_4$ is lost or wasted during post-harvest handling, storage, or processing. Of the consumed biomass ($M_3$), a share $c$ is directly used for human consumption ($M_D = cM_3$), and the remainder $(1-c)M_3 = M_B$ becomes by-products. A fraction $d$ of those by-products is re-utilised for food purposes ($M_R = dM_B$). This framework allows for the calculation of key utilisation rates, such as the gross utilisation rate ($M_3/M_H$) and net utilisation rate ($f = (M_D + M_R)/M_H$), to assess the efficiency of the fish value chain in delivering edible biomass to consumers.

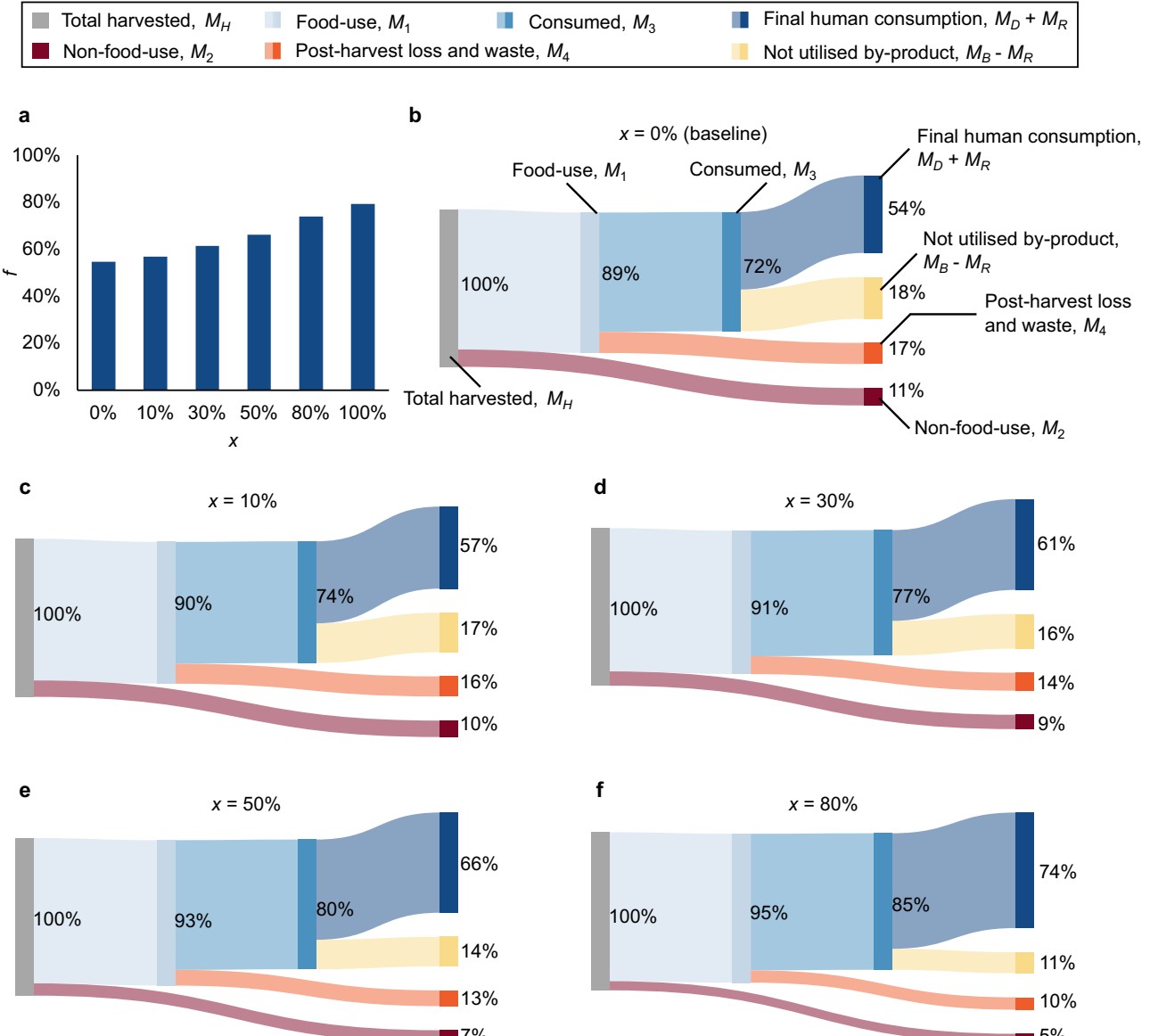

**Fig. 2 | Global transformation of fish-to-food conversion efficiency under increasing technology adoption. a** Net utilisation rate. Modelled net utilisation rate ($f$), defined as the percentage of harvested fish finally used for human consumption, at the global scale under technology adoption levels ($x$) of 0% (baseline), 10%, 20%, 50%, and 80%. **b–f** Material flow transformation. Sankey diagrams depicting the flow of harvested fish through the global food system under each scenario. Flows represent the share directed to food use, the proportion of consumed mass within that share, the fraction of edible material excluding by-products, and the extent of by-product re-utilisation. These diagrams illustrate how technological adoption reshapes the efficiency and structure of global post-harvest fish utilisation.

to a potential direct consumption level of 74% at 80% technology adoption (Fig. 2f). The 74% potential represents a theoretical upper bound under a harmonised, globally comparable adoption scenario. In practice, effectiveness is heterogeneous: cold chain acts primarily on early spoilage, handling on landing/transport damage, smoking/drying on processing/storage, and valorisation on by-product use. Our adoption parameter, therefore, standardises proportional improvements to enable comparison across regions with different chain geometries; it does not imply identical absolute reductions at every node or any reallocation away from reduction pathways.

In developed regions such as Europe and North America, where modern refrigeration and processing infrastructure is standard, spoilage losses in the distribution chain remain relatively low, typically under 10%[3]. In contrast, developing tropical regions experience substantially higher losses, with some FAO estimates suggesting they may reach up to 40% in certain regions of sub-Saharan Africa[3]. These losses predominantly affect small-scale fisheries due to limited ice availability, delayed marketing, and rudimentary processing methods. In South and Southeast Asia, artisanal supply chains experience elevated losses, often ranging between 15–30%[11,13,14]. For example, studies in Bangladesh and India have documented losses of 20–25%, mainly due to glut catches, lack of cold storage, and pest damage during sun-drying[11]. In Latin America, the situation is more variable: industrial, export-oriented fisheries tend to have lower losses, whereas remote coastal communities without access to ice continue to face considerable spoilage[15].

In China, the world's largest seafood producer, PHL is highly variable[16]. Industrial aquaculture and coastal fisheries benefit from relatively advanced infrastructure and cold chain coverage, with estimated spoilage losses typically below 12%[16]. However, losses can rise to 20–30% in inland and small-scale operations, particularly in western and rural provinces where access to ice, refrigeration, and efficient transport is limited[16,17].

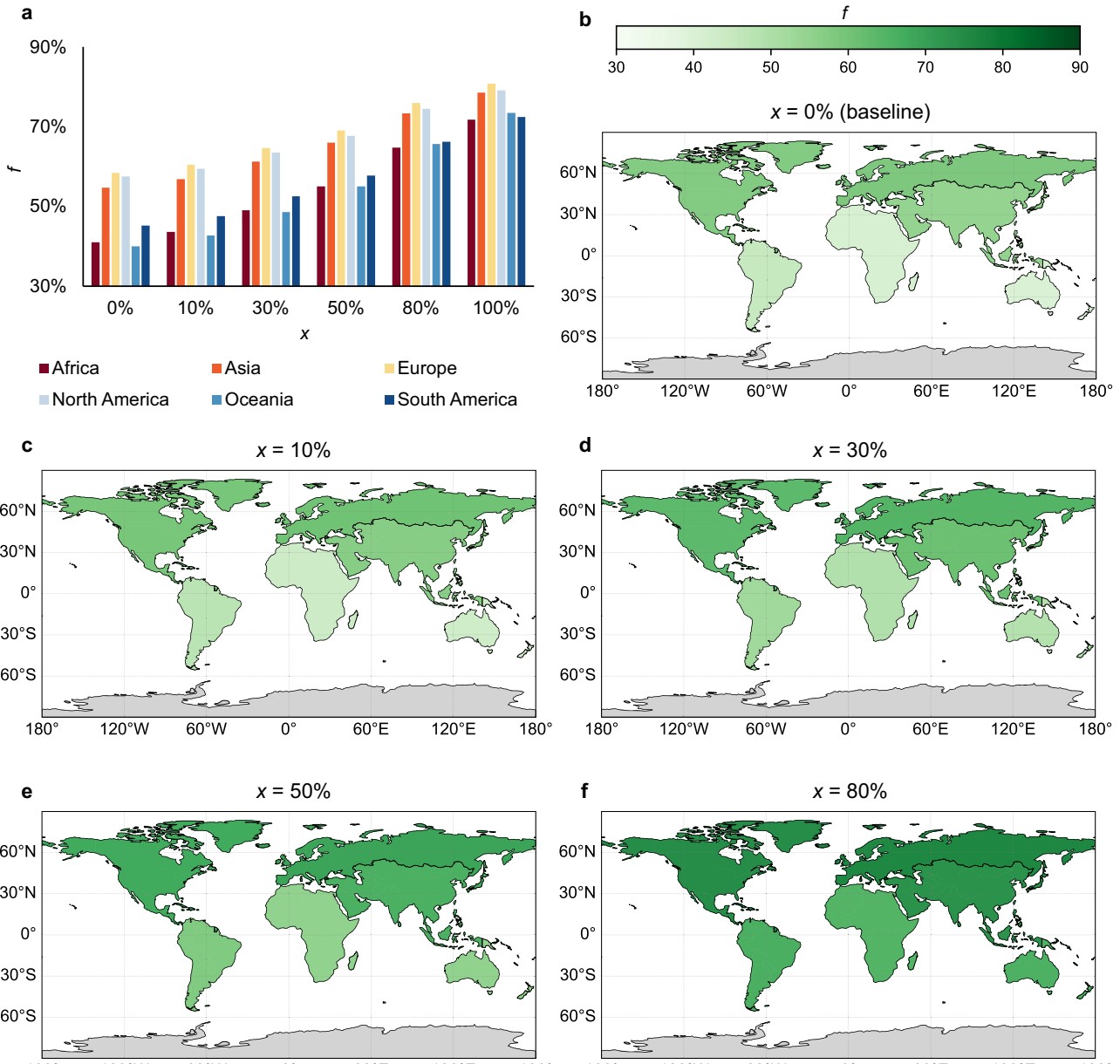

**Fig. 3 | Regional dynamics of fish utilisation under varying levels of technology adoption. a** Net utilisation rate. Modeled net utilisation rate ($f$), defined as the percentage of harvested fish finally used for human consumption, across five levels of technology adoption levels ($x$): 0% (baseline), 10%, 20%, 50%, and 80%. **b–f** Spatial distribution. Global maps show the regional variation in net utilisation rate ($f$) under each corresponding adoption level, highlighting spatial disparities in baseline efficiency and the localised impacts of technological interventions.

Notably, Oceania, despite its relative wealth, exhibits surprisingly low direct fish consumption rates. This paradox arises primarily from the region's substantial fish exports (over 50%[12] of harvested fish biomass is exported, compared to a global average of 38%[1]), a focus on non-food products such as fishmeal and fish oil, and the geographical dispersion of island communities, which complicates the logistics of local fish distribution and consumption and effective storage[1]. Consequently, roughly half of Oceania's harvested fish biomass ultimately does not enter local food markets[12]. Additionally, a structural divergence in dietary protein sources further contributes to this pattern: while developed countries, *i.e.*, Australia and New Zealand, rely predominantly on livestock products for protein, many Pacific Island nations derive over 50–90% of their animal protein intake from fish[18]. However, the latter often lack adequate food processing and cold chain infrastructure, leading to PHL and limited integration of fish into formal food systems.

### Technological innovations for post-harvest loss reduction

A diverse set of technological innovations is available to reduce PHL, targeting key loss points: maintaining the cold chain to prevent spoilage, improving drying and smoking techniques to extend shelf life, streamlining processing and handling to avoid physical losses, and converting what would otherwise be waste into useful products. Table 1 summarises the major types of innovations, real-world examples, and their demonstrated benefits in reducing losses.

Quantitative modelling revealed that combining interventions targeting handling ($a$), preservation ($b$), and by-product utilisation ($d$) produces a compounding benefit in utilisation efficiency. A hypothetical adoption rate of 50% across various technologies increased the fish net utilisation rate ($f$) from 54% to 66%. When adoption reached 80%, $f$ approached 74%, indicating a substantial reduction in losses and improved conversion of harvested fish into food products.

**Table 1 | Post-harvest loss reduction technologies and region-specific examples**

| Innovation category | Examples and regions | Loss reduction and benefits |
|---|---|---|
| Cold chain improvements | *Iceboxes and cold storage:* Distributed ice boxes for artisanal fishers in tropical regions (e.g., China[10,91], India[30–32], and East Africa[92]); installation of village freezing units (solar powered) in remote coastal communities (e.g., Peru[93,94], Persian Gulf[95–97], and Pacific islands[98]). | **Reduces spoilage** by keeping fish chilled from capture to market[99–102]. In India, providing small-scale fishers with iceboxes kept fish fresh beyond 8 hours and increased incomes ~ 20% by allowing sales at full price[30–32]. In China, portable insulated containers are widely used among coastal small-scale fishers to preserve freshness beyond 12 hours[10,91,99]. In Somalia, FAO-built cold rooms enabled a cooperative to export 10 t/month of fresh fish to regional markets, dramatically lowering local spoilage[92,103]. In Peru, solar-powered community freezers in Piura helped artisanal fishers transport fresh catch to Lima[93,94]. In Oman, mobile solar cold storage units reduced spoilage and expanded market access for remote villages[95–97]. These cold chain improvements enhance food safety, extend market reach, raise incomes, and reduce post-harvest losses for both small- and large-scale fishers[99,100]. |
| Drying and smoking technologies | *Improved ovens:* The Chorkor and Ahotor smoking kilns (e.g., West Africa[20–22] and Mexico[104]). *Solar Dryers:* Tent-style solar dryers (e.g., East Africa[8,33]) and solar chimney dryers (e.g., Bangladesh[105–107]). | **Prevents insect damage and oxidation**[45,108,109]. In Ghana, the Chorkor oven (introduced 1969) and newer Ahotor oven produce more uniform smoked fish with 40% less fuel and lower carcinogens, yielding longer shelf life and less breakage[20–22]. In Mexico, improved wood-fired smoking ovens reduced fuel use and PAHs for smoked mackerel[104]. In Malawi, a solar tent dryer (15×8 m) for small fish cut drying time, enabling all-weather drying and reducing losses to near zero[33]. In Bangladesh, solar chimney dryers keep products hygienic (no dirt/flies) without needing pesticides[105–107]. Dried fish output has higher quality and fetches better prices, incentivizing fishers to preserve excess catch rather than letting it spoil[45,108]. |
| Efficient handling and processing | *Auto-sorting and Improved Handling:* Conveyors and grading machines on processing lines, reducing manual damage, which is common in industrial fisheries (e.g., Europe[110–112], North America[113,114], and Japan[115–117]). *Packaging:* Low-cost plastic crates replacing jute sacks or heaping (Southeast Asia[118–121]). | **Reduces bruising, crushing, and waste**[114,118]. On large trawlers (Sweden and Norway[110–112]) and in processing plants (USA[113,114]), automated sorting minimises fish flesh damage and speeds processing. In Japan, companies like Asahi Machine Co. and Nikko have developed automated conveyor and grading systems for fish processing plants, reducing manual handling and improving product quality[115–117]. Proper containers prevent fish from being trampled or smashed during transport. These measures have cut physical handling losses (fish dropped or spoiled due to injury) from ~ 5% to under 1% in some Southeast Asian supply chains[118–121]. In small-scale contexts, even using baskets instead of piling fish on the floor can improve quality and yield[114,116,118]. |
| Fish by-product utilisation | *Valorisation of Off-cuts:* Fish heads, frames, skin, and viscera converted to fishmeal, fish oil, or even value-added human food products like fish sausages and stock. Notable in Iceland (cod)[122–124], Alaska, USA (pollock)[125–127], and Sweden (salmon and herring)[128–130], and developing countries (e.g., fish bone powder in West Africa[131]). | **Transforms would-be waste into value**[7,132–134]. Globally, 30–70% of fish biomass can be by-products depending on species[4,132]. Instead of dumping these, they are processed: ~ 30% of the world's fishmeal and 51% of fish oil now comes from such by-products[7,132,134]. In Iceland, technological advances over two decades enabled utilisation of cod heads, skin, and entrails, increasing total product yield by about 20%[7,122–124]. In Sweden, enterprises are experimenting with natural antioxidants to produce stable, high-quality protein isolates from fish frames for human consumption[7,128,129]. Meanwhile, in West Africa, small enterprises are turning smoked fish trimmings into soup bases and powder, creating new food items and income while reducing waste[131,135]. This circular approach means fewer fish need to be caught to obtain the same amount of edible output[7,132–134]. |

Field-based case studies support the modelled outcomes: improved post-harvest practices can significantly raise utilisation rates, particularly in regions with lower baseline performance. For example, if 50% of fishers in tropical Africa adopted better preservation technologies, regional gross utilisation rates could rise from 60–70% to over 80%. Empirical studies support this potential: a trial in Zambia using enhanced handling and solar drying in Lake Tanganyika fisheries reduced spoilage from approximately 15–30%[8,19]. In Ghana, the construction of over 500 improved Ahotor smoking ovens enabled women fish processors to produce smoked fish that remains mould-free for longer periods[20–22]. This improvement facilitated product transport to distant markets, reducing unsold or spoiled fish.

The economic cost of improving post-harvest fish preservation is relatively modest and potentially cost-effective. For instance, in Malawi, a per-fisher investment of solar tent dryers (FSTDs), which are often shared among 3–10 users, can be approximately 55–170 USD[23–26]. Willingness-to-pay studies show that fishers are prepared to contribute around 29 USD on average, suggesting that modest subsidies or cooperative ownership models could facilitate broader adoption[25,26]. Based on a scenario of 80,000 fishers (~ 50% of fishers in Malawi[27]) adopting these technologies, the total initial investment would range from approximately 4.4–14.0 MUSD. In India, similar low-cost solar dryers cost as little as ~24 USD, with a payback period of just 2–3 months due to reduced spoilage and improved market value[28,29].

## Nutritional and public health benefits

Reducing PHL in fisheries constitutes a highly effective strategy for improving global food and nutrition security. Each year, an estimated 31 Mt of fish (~ 17% of total harvested mass) are directly lost post-harvest (Fig. 2b, **red sandkey line**), representing a substantial missed opportunity to deliver essential nutrients. If recovered, these losses could yield approximately 850 million additional 100 g portions per day, enough to feed 10% of the global population 50% of their daily protein requirement, along with key micronutrients such as vitamin D, iodine, selenium, and long-chain omega-3 fatty acids (EPA and DHA)[7]. Assuming an average muscle protein content of 20%, this equates to roughly 2 Mt of high-quality protein, sufficient to meet the annual protein requirements of approximately 114 million adults (based on 17.5 kg per year, derived from 0.83 g·kg$^{-1}$·day$^{-1}$ for a 58-kg reference woman).

**Table 2 | Nutritional composition of fish protein isolates derived from processing by-products (per 100 g dry weight), compared with recommended dietary allowances (RDA) for healthy adults**

| Nutrient Category | Specific Nutrient | RDA† | Composition in fish protein isolate (per 100 g dry weight) | % RDA Coverage‡ |
|---|---|---|---|---|
| Macronutrients | Protein | 0.83 g·kg⁻¹·day⁻¹ ≈ 48 g·day⁻¹ @ 58 kg | 80.6 g | 168% |
| | Total lipids | 15–30%E (≈44–78 g·day⁻¹ @ 2000 kcal) | 12.5 g | / |
| | EPA + DHA | 250 mg·day⁻¹ | 1.6 g | 640% |
| Essential Amino Acids | Lysine | 30 mg·kg⁻¹·day⁻¹ ≈ 1.74 g·day⁻¹ | 6.4 g | 368% |
| | Histidine | 10 mg·kg⁻¹·day⁻¹ ≈ 0.58 g·day⁻¹ | 2.8 g | 482% |
| | Isoleucine | 20 mg·kg⁻¹·day⁻¹ ≈ 1.16 g·day⁻¹ | 3.8 g | 325% |
| | Leucine | 39 mg·kg⁻¹·day⁻¹ ≈ 2.26 g·day⁻¹ | 5.9 g | 263% |
| | Methionine | 15 mg·kg⁻¹·day⁻¹ ≈ 0.87 g·day⁻¹ | 2.6 g | 295% |
| | Phenylalanine | 25 mg·kg⁻¹·day⁻¹ ≈ 1.45 g·day⁻¹ | 3.3 g | 226% |
| | Threonine | 15 mg·kg⁻¹·day⁻¹ ≈ 0.87 g·day⁻¹ | 3.5 g | 402% |
| | Tryptophan | 4 mg·kg⁻¹·day⁻¹ ≈ 0.23 g·day⁻¹ | 1.0 g | 431% |
| | Valine | 26 mg·kg⁻¹·day⁻¹ ≈ 1.51 g·day⁻¹ | 3.9 g | 259% |
| Minerals | Selenium | 26 µg·day⁻¹ | 23.4 µg | 90% |
| | Calcium | 1 000 mg·day⁻¹ | 29.3 mg | 3% |
| | Magnesium | 310 mg·day⁻¹ | 19.8 mg | 6% |
| Vitamin | Vitamin D | 15 µg·day⁻¹ | 5.0 µg | 33% |

†RDA values refer to a non-pregnant adult woman (19–30 y, 58 kg), based on WHO/FAO (2001–2007); Codex NRVs and NAM DRIs were used where global benchmarks were not available. Amino acid needs are converted from mg·kg⁻¹·day⁻¹.

‡% RDA = (Nutrient content ÷ Daily requirement) × 100. Total lipids are expressed as % energy and excluded from %RDA coverage due to variation in individual energy needs.

To evaluate the nutritional potential of recoverable loss streams, protein isolates were benchmarked from fish by-products using pH-shift processing. As shown in Table 2, the resulting isolate contained 80.6 g of protein and 1.6 g of EPA + DHA per 100 g dry weight, with total lipids at 12.5 g and negligible ash. A single 100-g portion delivered 168% of the total daily protein requirement and 226–482% of the requirement for individual indispensable amino acids, including a fourfold sufficiency in lysine and histidine. EPA + DHA content exceeded 640% of the recommended daily intake (250 mg/day), while selenium and vitamin D contributed 90% and 33% of daily requirements, respectively. The residual mineral fraction also supplied trace amounts of calcium and magnesium.

Given that fish frames, heads, and viscera constitute 40–60% of landed biomass, the ≥80% protein recovery rates demonstrated here underscore the potential to convert these undervalued fractions into microbiologically safe, nutrient-dense ingredients[7]. These isolates offer broad functional utility, including fortification of staple foods and formulation of shelf-stable protein powders[7]. The convergence of increased edible yield and exceptional nutrient density positions PHL reduction and biomass valorisation as high-impact interventions for enhancing dietary quality and resource efficiency in aquatic food systems.

**Economic benefits**

PHL reduction initiatives in fisheries have demonstrated measurable economic and environmental benefits across various regions. For example, in India, a government-led intervention that provided iceboxes to local fish vendors resulted in an average income increase of approximately 20% during the trial period, attributed to better preservation and improved fish quality[30–32]. However, the discontinuation of the program and reduced access to iceboxes caused vendor income to regress, underscoring the importance of such technologies.

A modelling framework developed in this study assessed the economic efficiency of improved utilisation. Two factors were introduced in the model: the percentage of supply chain costs in the final market value ($k_1$) and the percentage of fixed costs in the total supply chain costs ($k_2$). Positive, neutral, and negative predictions are made based on various values of $k_1$ and $k_2$, as indicated in Fig. 4. As the cost saving in the supply chain per unit mass is in inverse proportion to $f$ (derived in Methods), a small level of technology

adoption, e.g., 10%, can result in a high cost and price reduction. Based on the neutral prediction, by increasing the level of technology adoption from 0% to 80%, the cost per tonne of fish marketed dropped by 374 USD (from 4479 USD to 4105 USD), primarily due to the distribution of fixed costs (such as boats, fuel, and processing facilities) over a greater volume of saleable fish. Importantly, this cost efficiency translated into a total market price reduction of 748 USD/t, an 8.4% decrease from the original market price. This price drop represents a substantial economic benefit. It should be noted that the price effects reported here reflect efficiency gains conditional on successful technology adoption and do not incorporate capital, operating, or transition costs associated with implementation. This analysis also does not account for the system-wide improvements at a regional and country level that would need to occur in order for the electrical grid to be stable enough for improvements in energy-intensive technology improvements such as cold chain storage.

Beyond economic gains, technological innovations also yielded environmental benefits: in Ghana, improved smoke ovens like the Morrison and Ahotor models used 40% less firewood, reducing deforestation[20–22], while in Malawi, solar tent dryers offered energy- and time-efficient alternatives to traditional smoking methods[33].

**Discussion**

Our study highlights pervasive inefficiencies within aquatic food production, where globally only 54% of harvested fish biomass is directly utilised for human food, with 17% lost as waste post-harvest, and the remaining 28% split across non-food use and unutilised by-products. These losses are not uniform: low-income regions often experience higher post-harvest spoilage due to inadequate infrastructure, whereas wealthier regions see more waste at the retail and consumer stages. For example, sub-Saharan Africa and parts of South Asia bear the highest post-harvest loss burdens – up to 40% - due to infrastructural and capacity constraints. However, we demonstrate that a coordinated portfolio of technological interventions could increase net utilisation in direct human consumption to over 85%, equating to more than 850 million additional daily servings of nutrient-dense fish. These "hidden harvests" could provide a low-footprint strategy to meet rising protein and micronutrient demands, particularly in food-insecure regions, without additional extractive pressure on aquatic

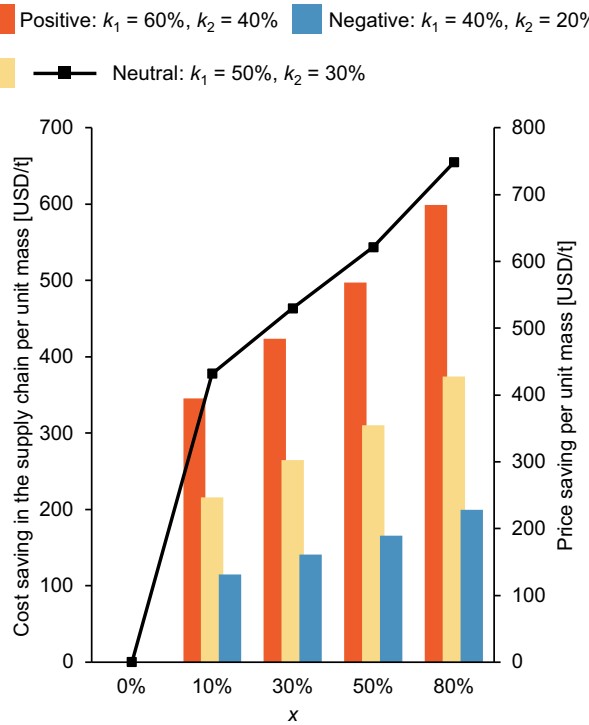

**Fig. 4 | Economic benefits of technology adoption in the fish supply chain.** Bar plot: Cost saving. Modeled cost savings ($\triangle \left( \frac{C}{M_F} \right)$) [USD/t] under technology adoption levels ($x$) of 0% (baseline), 10%, 20%, 50%, and 80%. Line plot: Cost saving. Modeled cost savings in the supply chain per unit mass of harvested fish ($\triangle P_F$) [USD/t] under technology adoption levels ($x$) of 0% (baseline), 10%, 20%, 50%, and 80%. The variations in $\frac{C}{M_F}$ and $P_F$ reflect reductions in processing losses and improved resource efficiency.

ecosystems. Our findings reinforce global assessments by the FAO[1], and synthesise with global calls[6,34–36] to enhance the efficiency of aquatic food systems as a central pillar of future food security. This emphasis echoes the FAO and other experts who argue that cutting post-harvest losses is one of the most effective ways to increase seafood availability without further stressing overfished stocks[37].

We demonstrate that technological interventions can serve as pivotal leverage points in transforming aquatic value chains, not merely by addressing losses but by unlocking systemic efficiencies that span preservation, processing, and value creation. Table 1 quantifies the discrete impacts of interventions, and the broader literature highlights their transformative role in reshaping value chain dynamics, particularly in under-resourced geographies. Cold-chain innovations, for example, deliver benefits that extend well beyond spoilage reduction, stabilizing supply, improving sensory attributes, and enabling longer distribution networks that support access to higher-value markets and reduce income volatility for fishers[38–41]. These systems also reduce microbial contamination, a key determinant of both food safety and consumer trust, particularly in informal markets[42], and improved handling and storage practices further cut losses while enhancing delivery reliability. Collectively, addressing these foundational inefficiencies allows the sector to deliver more nutrition from the same catch, consistent with recent meta-analyses indicating that 30–50% of post-harvest losses could be avoided through improved practices[43].

Beyond cold storage, our work highlights the importance of improved processing technologies – findings that are supported by global experiences with introducing such innovations. For instance, enhanced drying and smoking innovations like the Ahotor oven and solar tunnel dryers provide energy-efficient alternatives that improve product safety and better preserve micronutrient integrity compared with traditional techniques, thereby

increasing the nutritional value of processed products[44]. They also reduce exposure to carcinogenic chemicals—an often-overlooked occupational hazard for processors[45,46]. Additionally, their modular and locally adaptable design makes them particularly suitable for artisanal contexts, where cost and gender-sensitive labour dynamics are critical[47]. Women processors in Ghana using the Ahotor oven report not only a drop in fuel costs but also faster processing times and healthier working conditions, with one initiative documenting a reduction from five days to under two days to smoke a given quantity of fish[48]. Such improvements validate our assumption that better preservation technology can mitigate losses while delivering co-benefits (in this case, labour efficiency, cost savings, and health protection). Likewise, solar drying innovations are proving transformative in other regions. In Cambodia, for example, the introduction of solar fish dryers has led to higher-quality dried fish and greater incomes for small processors, compared to traditional sun-drying on the ground[49]. A recent study found that processors using solar dryers produced more dried fish at higher selling prices, because the product was cleaner and less prone to spoilage, resulting in better livelihoods for fishing communities[49].

Improved handling approaches are a further technological innovation that can markedly reduce PHL. The importance of careful handling during harvest is already established in the literature, where small details such as fishing hook design have a large influence on potential damage during harvest[50]. Table 1 explains how automated conveyor and grading systems reduce manual handling and reduce the chance of fish bruising post-harvest. Proper container systems further reduce bruising, drawing parallels to the arable sector, where correct packaging is of critical importance[51]. Together, these techniques also minimise the time between fish capture and packaging, which, as demonstrated by the salmon industry, is a key determining factor in maintaining fish quality[52].

Valorisation of by-products holds arguably the most scalable potential for high-impact change. Globally, fish side streams represent a vast reservoir of only partially tapped nutrition and economic value[6,36,53]. Table 1 demonstrates how fish by-products such as bones and frames, previously only of value to non-human food sectors, can be valorised into high-value human food products such as protein powders and bone powders. In this study, protein isolate is used as an illustrative example of a by-product–to–food pathway because it allows for globally comparable modelling. However, by-products can enter many other food applications, including mechanically recovered meat, collagen- and gelatin-based ingredients, fermented products, oils for direct consumption, and traditional regional dishes[54], which together represent a substantially larger utilisation potential than quantified here. As a result, our estimates should be interpreted as conservative lower bounds on the true human-food potential of by-products. The recovered protein ingredients not only meet WHO benchmarks for amino acid profiles but also exhibit functional properties—such as emulsification and solubility—suitable for potential incorporation into school meals, emergency nutrition programs, and processed foods in global regions where need is greatest[55,56]. Combined with further encouragement for direct human consumption of by-products in traditional recipes, this could make a major impact on global nutritional security[57].

Crucially, our analysis demonstrates that these interventions are not isolated silos but complementary and synergistic when implemented together – a systems perspective supported by other holistic studies of food loss. Improvements in cold chain, processing, and by-product reuse tend to reinforce each other's benefits. For example, if better cold storage prevents spoilage, more raw material remains available to be turned into products (including those made from by-products); conversely, finding edible uses for fish off-cuts increases the effective yield from each kilogram caught, making investments in preservation more worthwhile. Our model showed that when interventions are deployed in tandem at high adoption levels, the net efficiency gains compound, resulting in over 80% of harvested fish being utilized as food. This finding resonates with the "full value chain" approach advocated in recent global reports, which argue that only a coordinated strategy can unlock the maximum gains from aquatic foods. In other words,

tackling one part of the problem (e.g., cold-chain gaps) is important, but tackling multiple stages simultaneously – from the moment of catch through processing and distribution – yields multiplicative benefits. This insight is in line with systems-level analyses in the Blue Food Assessment and others, which note that fragmented efforts would not capture the full potential reduction in losses[6,58,59]. Our study, therefore, not only quantifies these synergistic effects but also underlines a key point of convergence with existing evidence: integrated interventions across the supply chain are far more effective than piecemeal fixes, reinforcing calls for comprehensive policy packages to improve food system efficiency.

The nutritional and public health implications of reducing PHL are substantial. Fish are uniquely positioned within the global food system due to their density of high-quality protein and bioavailable micronutrients, including omega-3 fatty acids, selenium, vitamin D, and iron[60]. The reduction of PHL can vastly increase the delivery of these essential nutrients to underserved populations. Table 2 demonstrates the remarkable nutritional density of valorised fish by-products, which offer not only macronutrient recovery but also substantial micronutrient contributions.

Of particular note, fish protein isolates derived from by-products contain high concentrations of lysine, threonine, and histidine - amino acids often lacking in plant-based diets - and meet or exceed the WHO recommended nutrient intake levels[61–64]. These products can be incorporated into culturally acceptable, shelf-stable, staple foods that could be especially effective in addressing malnutrition in children and those in greatest need[55,56,65]. Moreover, reducing post-harvest inefficiencies aligns directly with SDG 2 (Zero Hunger), and advances multiple WHO Global Nutrition Targets, particularly in addressing stunting, anaemia, and low birth weight[63,64]. Our data support calls that aquatic foods should be central to global nutrition policy, especially as climate-resilient alternatives to terrestrial animal-sourced foods[66].

Reducing PHL in fish value chains carries significant economic and livelihood implications. Our economic modelling indicates that reducing loss, by taking technology adoption from 0–80%, leads to an 8.4% reduction in the market price of fish to the consumer whilst also reducing costs in the supply chain, not only increasing consumer accessibility to affordable, nutritious food, but also creating new economic opportunities for fisherfolk and processors.

Case studies support these conclusions. When deployed in Cambodia, the aforementioned solar dryers enable the creation of a higher quality fish product that delivers more income to fish producers and sellers whilst increasing the supply of nutritious food to consumers[49]. There is an opportunity for new products and business development from greater utilisation of byproducts, ranging from canned salmon soup to minced carp meat as examples[67,68]. It is important to highlight that, without equitable policy design, benefits may accrue to better-capitalised actors, exacerbating inequality. Targeted subsidies, inclusive financing mechanisms, and gender-responsive training programs are critical to ensure that women and small-scale fishers—who represent much of post-harvest labour—are not excluded[69,70].

Enhancing post-harvest efficiency offers clear environmental benefits and supports long-term sustainability. From an ecological perspective, increasing edible yield derived from each fish reduces fishing pressure and contributes to more sustainable resource use. By increasing the proportion of harvested biomass that reaches the consumer, fewer fish are needed to meet each unit of demand, aligning with sustainable intensification strategies and blue food system principles[1,6,36]. Additionally, by enabling an increased total delivery of fish-based protein to the human population, post-harvest value chain improvements could enable fish consumption to displace consumption of meat-based proteins. Fish production systems have markedly lower carbon footprints than meat production systems[66], and also come with lower land and freshwater footprints[71], creating environmentally positive opportunities.

Additionally, creating a "wave of change" in preferences is feasible when product form, price, and convenience match local tastes. Partnering with processors to reformulate small pelagics into familiar dishes,

using public procurement to seed demand, and running culturally grounded campaigns can normalise these foods within mainstream diets. Such measures are complements, not substitutes, for technical loss reduction.

Several recent empirical studies support the magnitude and drivers of PHL reflected in our model and suggest that interventions along handling, cold-chai,n and processing stages can substantially reduce losses. For example, a recent review of fisheries in Sub-Saharan Africa estimated that total PHL (inclusive of both physical wastage and quality/market losses) often reaches 30–40% of landings under current handling conditions[72,73]. A case study in Zambia found that while many artisanal fishers suffered notable PHL, improved preservation practices (e.g., timely chilling and smoking) consistently lowered total losses toward the lower end of regional ranges[8]. Moreover, recent field surveys in marine fisheries of Bangladesh estimate that post-harvest losses amount to 15–20% (physical + market losses combined), underlining that even in better-developed value chains, non-trivial PHL remains a constraint on food and nutrition security[74]. These findings reinforce our model assumptions: that a large proportion of harvested fish fails to reach consumers without appropriate handling and infrastructure; and that targeted interventions, such as cold storage, rapid processing, or by-product valorisation, could recover a meaningful share of this lost biomass. Incorporating such real-world data alongside our scenario analysis helps validate that the utilization improvements we project are both plausible and potentially impactful.

While this analysis provides robust, globally scalable estimates of best-case system-level potential, several limitations and areas for future research are acknowledged. First, we do not attempt to estimate the likelihood of achieving the outcomes reported here or to derive a single "realistic" global value that fully considers all realised costs and externalities. At a global scale, technology adoption in fisheries is shaped by heterogeneous and interacting factors, including governance capacity, access to capital and infrastructure, species composition, supply-chain organisation, regulatory constraints, and gendered labour structures. These drivers vary not only across regions but also across fisheries and value-chain nodes within the same country. As a result, adoption likelihood is not a well-defined global parameter: any single estimate would require strong assumptions about future policy, investment, and behavioural responses, and would risk conveying a false sense of precision. We therefore restrict the analysis to bounding what is technically and structurally feasible under harmonised adoption, and interpret the results as upper limits on recoverable potential rather than expected outcomes. A further structural constraint is the persistence of reduction fisheries (fishmeal, fish oil, and other non-food uses), which are institutionally entrenched and primarily shaped by market demand and industry configuration rather than post-harvest inefficiency. Consistent with this, our modelling applies technology adoption only to biomass already destined for direct human consumption and by-product reuse; it does not assume reallocation away from reduction sectors, and policy- or demand-side shifts that could alter the food/non-food split lie outside our scope. In addition, treating technological adoption as proportional and harmonised across value-chain nodes is a modelling choice that facilitates global comparability, but necessarily abstracts away from node-specific and context-dependent adoption dynamics. Assumptions regarding average regional loss rates, adoption dynamics, and nutrient recovery efficiencies may therefore not hold uniformly across species, geographies, or production systems. This choice enables system-level bounding analysis, but should not be interpreted as representative of real-world adoption patterns. Socio-behavioural drivers of uptake, including cultural acceptability, gender norms, and access to credit, also merit deeper investigation. Then, although innovations that recover value from streams typically counted as post-harvest loss (e.g., secondary processing of downgraded product) could further increase human-food availability, quantifying this pathway at a global scale is currently infeasible. Recovery potential depends strongly on loss mechanisms, species, timing, food-safety constraints, and local regulatory frameworks, and no harmonised

datasets exist to estimate feasible recovery yields without introducing double counting with by-product utilisation. For this reason, we treat post-harvest loss as unrecoverable within the system boundary and interpret our estimates as conservative with respect to such recovery pathways. Future work could narrow these gaps by integrating country- and species-weighted adoption scenarios, informed by empirical uptake data, governance conditions, and species-specific recovery potentials. Moreover, because the economic analysis focuses on efficiency gains conditional on adoption, it does not account for upfront investment requirements, financing constraints, or learning costs, which in practice may delay or partially offset short-term price effects, particularly for small-scale actors.

Future work should resolve heterogeneity by species, product form, chain node, and region (e.g., cold-chain elasticity at landing vs retail, species-specific dressing yields and by-product potentials, and contexts where reduction fisheries dominate). Priorities include longitudinal studies that track real-world adoption, environmental life-cycle assessments (LCAs) of post-harvest technologies, and species-specific nutrient profiling to inform public-health applications, alongside evaluations of social-equity outcomes—especially labour distribution, income control and access to decision-making.

In conclusion, unlocking the post-harvest potential of aquatic food systems offers a cost-effective, high-impact strategy for achieving global food security, advancing nutrition, and supporting inclusive economic growth. The technological solutions are proven, affordable, and scalable. What is now needed is political will, catalytic investment, and integrated policy frameworks to ensure widespread adoption. There is potential here to deliver an additional 850 million portions of fish per day and reduce cost to the consumer by nearly 10%, without catching any additional fish, if PHL is mitigated. This emphasises a fundamental point – the next step-change in sustainable blue food systems lies not in extracting more from aquatic environments, but in using what is already harvested more wisely. It also entails fully recognising the breadth of food applications enabled by by-products, well beyond the single illustrative pathway modelled here. Finally, to refine these estimates and strengthen their policy relevance, future work should incorporate species-weighted nutrient modelling with region-specific species baskets and dressing yields, allowing more granular assessments that reflect the diversity of fisheries and processing practices worldwide. By shifting focus to post-harvest efficiency and valorisation, we can realise a future in which nutritious, equitable, and climate-smart aquatic foods nourish billions.

## Methods

This study combined a quantitative modelling approach with a qualitative synthesis of case studies and literature. The aim was to estimate the impact of technological interventions on post-harvest fish loss reduction and to contextualise those findings with real-world data.

### Modelling fish utilisation and loss reduction

A mathematical model was developed to represent the fish post-harvest chain (Fig. 1). In this study, harvested fish refers specifically to fish that are landed and reported in FAO production statistics[1]. Fish discarded at sea are not included, as they do not enter the supply chain and therefore fall outside FAO's utilisation accounts. Accordingly, post-harvest losses in this analysis refer only to losses occurring after landing, excluding any at-sea discards. Crustaceans and molluscs are excluded. The model defines the rate of utilisation ($f$), as the proportion of harvested mass used as food, measuring the efficiency of transforming harvested fish for direct use as food. The value of $f$ is calculated by Eq. 1, where $M_F$ is the mass finally utilised as food, and $M_H$ is the total harvested mass. $M_F$ is the sum of the mass of direct human consumption ($M_D$) and re-utilised by-products ($M_R$). A rate of 100% indicates that the entire harvested fish mass is utilised as food, with no food loss and waste (FLW) or by-products, while a rate of 0% means that none of the harvested mass is converted into food. $f$ is influenced by three interconnected stages: food and non-food applications, post-harvest FLW, and

by-product management. These stages collectively influence the rate of utilisation.

$$f = \frac{M_F}{M_H} \times 100\% \tag{1}$$

To better represent intermediate utilisation stages along the fisheries value chain, we introduce four sub-rates: $a$ denotes the proportion of harvested fish entering food-use pathways; $b$ is the proportion of this food-use stream that remains available as edible fish products after all losses during capture, landing, transport and processing; $c$ the edible-portion yield obtained at processing; and $d$ the proportion of the resulting structural inedible by-products that are re-valorised into food. In this study, we follow the loss accounting hierarchy defined in the WEF Aquatic FLW Annex (2024)[4] while maintaining consistency with FAO's SOFIA 2024 utilisation categories[1]. Losses occurring during production and processing stages include both edible and inedible fractions that reduce the biomass available as edible fish products. These reductions are first captured in $(100\% - b)$, which therefore represents the total post-harvest reduction of edible-fish availability along the food-use pathway. In this model, biomass counted as post-harvest loss $(100\% - b)$ is treated as unrecoverable for direct human food within the system boundary; potential re-entry of degraded or discarded edible fractions via secondary recovery pathways is not explicitly modelled. Structural inedible mass (e.g., heads, frames, skin, viscera) is then explicitly partitioned from these reductions through the edible-portion parameter $c$, generating by-products that have the potential to re-enter the food system via $d$. Because each unit of biomass is either retained as edible fish, diverted to by-products, or lost from the edible pathway, all flows remain sequential and mutually exclusive, preventing any double-counting across categories. The combined net-utilisation rate of harvested fish for direct human consumption is therefore expressed as Eq. 2.

$$f = a \cdot b \cdot (c + (100\% - c) \cdot d) \tag{2}$$

The baseline condition, i.e., the current condition without extra technology adoption, is defined. The value of $f$ for the baseline condition is calculated by Eq. 3, where $a_0$, $b_0$, and $d_0$ are based literatures. The values of $a_0$, $b_0$, and $d_0$ varies across continents which are listed in Supplementary Table 1, while globally, $a_0 = 89\%$ [1,4], $b_0 = 81\%$ [4], and $d_0 = 30\%$ [4]. The value of $c = 65\%$ [75] is considered a constant throughout the analysis. It should be mentioned that utilisation and loss metrics are already reported at global and continental scales[1,4], so no country-level aggregation or re-averaging was performed in this study.

$$f_0 = a_0 \cdot b_0 \cdot \left(c + (100\% - c) \cdot d_0\right) \tag{3}$$

By giving various values of the levels of technology adoption ($x$), the improved $a$, $b$, and $c$ can be calculated by the Equations Eqs. 4–6, $\lim_a$ is the maximum achievable improvement in the food-allocation stream (e.g., a fraction of non-food uses is structurally non-recoverable), $\lim_b$ is the maximum stabilisable percentage of losses along the food-use pathway, e.g., freshly eaten fish is excluded, $d_{max} = 70\%$ is the maximum percentage of by-products that can be used for food (e.g., bones are excluded. The value of $\lim_a$ varies across continents, which are listed in Supplementary Table 1, while globally, $\lim_a = 68\%$ [4], $\lim_b = 56\%$ [76]. $d_{max} = 70\%$ [77] is regarded as a global constant.

$$a = a_0 + \left(100\% - a_0\right) \cdot a_{lim} \cdot x \tag{4}$$

$$b = 100\% - \left(100\% - b_0\right) \cdot \left(100\% - b_{lim} \cdot x\right) \tag{5}$$

$$d = d_0 + \left(d_{lim} - d_0\right) \cdot x \tag{6}$$

## Nutrition analysis

Throughout this study, a portion is defined as 100 grams of edible fish, following serving-size conventions used by FAO/WHO nutritional guidelines[63,64]. All estimates of additional availability are expressed in 100-gram edible portions unless otherwise noted.

To evaluate the nutritional adequacy of fish processing by-products for human consumption, we used the composition of protein isolates derived from Atlantic herring (*Clupea harengus*) backbones as an illustrative case. Herring backbones are a widely available and underutilised by-product in Nordic pelagic fisheries. Because oily pelagics such as herring typically exhibit richer fatty acid and micronutrient profiles than many lean whitefish and numerous freshwater taxa, we treated the herring-based values as an upper bound for certain nutrients rather than a global average[78–81]. As such, realised nutrient recovery from by-products will depend on regional species composition and processing characteristics. Nutrient composition values (per 100 g dry weight, DW) were compiled from peer-reviewed studies describing pH-shift processed protein isolates from herring by-products[78–81]. This processing method enables high protein recovery while removing bones and lipids, resulting in a consistent, low-ash, and protein-rich ingredient. To ensure consistency, only data expressed on a dry-weight basis were used for calculations.

Nutrient requirements were standardised to a non-pregnant adult woman (19–30 years, 58 kg body weight). Recommended dietary allowances (RDAs) were primarily sourced from WHO/FAO expert consultations (2001–2007), which remain the most globally harmonised nutrient references[63,64]. Where global values were unavailable, Codex Nutrient Reference Values (NRVs)[82] or U.S. Dietary Reference Intakes (DRIs, 2022)[83] were used. Amino acid requirements expressed in $mg \cdot kg^{-1} \cdot day^{-1}$ were converted to absolute daily amounts. Fat intake was expressed as a percentage of dietary energy (%E), with approximate g/day conversions assuming a 2,000 kcal diet[84].

Percentage RDA coverage was calculated as Eq. 7. Total lipids were excluded from % coverage due to variation in individual energy requirements and their expression on an energy basis.

$$RDA \ Converage = \left( \frac{Nutrient \ content \ per \ 100 \ g \ DW}{Daily \ requirement} \right) \times 100\% \quad (7)$$

## Economic analysis

With the technology adoption, the improved $f$ leads to increased final utilised mass. A corresponding improvement in food mass ($\triangle M_F$) can be calculated by Eq. 8, where $M_{F0} = f_0 \cdot M_H$ is the final utilised mass at the baseline condition. Assume the effective increase in market supply $\triangle M_S = \triangle M_F$.

$$\triangle M_F = M_F - M_{F0} = M_H \cdot \left( f - f_0 \right) \quad (8)$$

The supply chain cost ($C$) comprises fixed ($C_f$), e.g., infrastructure and equipment, and variable ($C_v$) components, as in Eq. 9.

$$C = C_f + C_v \quad (9)$$

The economies of scale indicate that as more harvested fish are utilised, $C_f$ is spread over a larger volume of output, reducing the average cost per unit. Therefore, $C_f$ is considered a constant in this study. According to the marginal cost, increasing effective supply reduces the marginal cost of production if fixed resources are better utilised. Given the initial variable cost ($C_{v0}$), the improved supply chain cost can be calculated by Eq. 10.

$$C = C_f + \frac{f}{f_0} C_{v0} \quad (10)$$

In this context, the cost saving in the supply chain per unit mass ($\Delta \left( \frac{C}{M_F} \right)$) can be derived by Eq. 11, which is not dependent on $C_{v0}$.

$$\Delta \left( \frac{C}{M_F} \right) = \frac{C_0}{M_{F0}} - \frac{C}{M_F} = \frac{(f_1 - f_0) \cdot C_f}{f_1 \cdot f_0 \cdot M_H} \quad (11)$$

As of 2022, the global production of aquatic animals reached $M_H$=185 Mt, comprising 91 Mt from capture fisheries and 94 Mt from aquaculture[12]. Determining the total cost of the global fish supply chain is complex due to the numerous variables involved, including production, processing, transportation, and marketing expenses. While specific global figures are not readily available, the seafood industry is a significant economic sector, with global trade in fish and fish products valued at $V_T$=143 BUSD annually[1,85].

The IMF provides benchmark prices representative of the global market, determined by the largest exporter of a given commodity[86]. For instance, the global price of fish in 2023 was approximately $P_F$ = 8,957.45 USD/t [87]. The market value can then be estimated as $V_F$ = 1.25 TUSD annually.

Based on industry reports, the supply chain typically consumes $k_1 \sim 40 - 60\%$ of the final market value, depending on the fish type and supply chain efficiency[88]. Hence, the total cost of the supply chain can be estimated as $C_0 = k_1 \cdot V_F$. Meanwhile, industry benchmarks suggest that fixed costs account for $k_2 \sim 20 - 30\%$ of total supply chain costs[89]. Hence, the total cost of the supply chain can be estimated as $C_f = k_2 \cdot C_0$.

## Case study data collection

To enrich and ground our quantitative findings, data from academic papers, technical reports, and organizational case studies on fish post-harvest loss interventions were collected. Targeted literature searches using keywords like "fish post-harvest loss reduction," "fish cold chain case study," "improved fish smoking impact," and region-specific terms (e.g., "post-harvest fish Africa losses") were performed. Sources included peer-reviewed journals (e.g., *World Development* meta-analysis, *Agricultural Research* case in Malawi), FAO reports, WorldFish/NGO blogs, and press releases. Preference was given to sources with quantitative estimates of loss percentages or outcomes of interventions (income changes, percentages of loss reduction, etc.). For example, from an article on India, the statistic of 15,000 INR crores lost to fish post-harvest issues and a 20% income gain from providing ice infrastructure were obtained[5]. To support and ground our discussion of post-harvest loss interventions, we drew on peer-reviewed and public sources that provide quantitative and qualitative evidence. For example, a recent review of Sub-Saharan African fisheries reports that PHL interventions across capture, handling, and processing stages could substantially narrow the gap between supply and demand[72]. We also consulted an NRI scoping report summarising loss-reduction initiatives and cold-chain improvements in multiple African countries[90]. These findings were synthesised thematically (cold chain interventions, drying/smoking, by-product use, etc.), which informed the structure of Table 1 and the examples cited in **Results**.

All citations of numeric data or specific claims in the text correspond to the original sources. Those references were maintained in References for transparency and to allow readers to consult the original studies or reports for more detail.

## Data availability

Source data are provided with this paper. All data are available at https://github.com/jnzhangfoodscience/fish-postharvest-loss.git.

## Code availability

Source code for data processing and visualization is available at https://github.com/jnzhangfoodscience/fish-postharvest-loss.git.

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

## Acknowledgements
D.F.W. was funded by a Henslow Fellowship at Murray Edwards College, University of Cambridge, and a BBSRC Impact Accelerator Grant G116601. The other authors received no external funding for this work.

## Author contributions
All authors contributed to the study conception and design. J.Z., H.Z., and O.P. led the data collection and data analysis. H.W. and D.F.W. led the writing of the manuscript. All authors reviewed the manuscript before submission.

## Competing interests
The authors declare no competing interests.
