## [Transparent Peer Review file · Communications Sustainability]

Technology-driven reduction of fish post-harvest loss could enhance food security and economic resilience

Corresponding Author: Dr David Willer

Version 0:

Decision Letter:

Dear Dr Willer,

Please accept our apologies for the delay in sending a decision on your manuscript. Your manuscript titled "Technological innovations can reduce post-harvest fish losses and sustainably improve nutritional and economic outcomes" has now been seen by 2 reviewers, and we include their comments at the end of this message. They find your work of interest, but some important points are raised. We are interested in the possibility of publishing your study in Communications Sustainability, but would like to consider your responses to these concerns and assess a revised manuscript before we make a final decision on publication.

We therefore invite you to revise and resubmit your manuscript, along with a point-by-point response that takes into account the points raised. Please highlight all changes in the manuscript text file.

Please submit your point-by-point responses as a separate file, distinct from your cover letter where you can add responses to the Editors' comments that you do not want to be made available to the reviewers. Word files are preferred. We recommend that any figures, tables or graphs that are included in the response to reviewers are also included in the main article or Supplementary Information.

Please use the following link to submit your revised manuscript, point-by-point response to the reviewers' comments (which should be in a separate document to any cover letter), a tracked-changes version of the manuscript (as a PDF file) and the completed checklist:

Link Redacted

We hope to receive your revised paper within six weeks; please let us know if you aren't able to submit it within this time so that we can discuss how best to proceed. If we don't hear from you, and the revision process takes significantly longer, we may close your file. In this event, we will still be happy to reconsider your paper at a later date, as long as nothing similar has been accepted for publication at Communications Sustainability or published elsewhere in the meantime.

Please do not hesitate to contact us if you have any questions or would like to discuss these revisions further. We look forward to seeing the revised manuscript and thank you for the opportunity to review your work.

Best regards,

Alice Drinkwater, PhD
Consulting Editor

EDITORIAL POLICIES AND FORMATTING

- Behavioural and social science
- Ecological, evolutionary & environmental sciences
- Life sciences

Furthermore, please align your manuscript with our format requirements, which are summarized on the following checklist: <https://www.nature.com/documents/commsj-phys-style-formatting-checklist-article.pdf> Communications Sustainability formatting checklist

and also in our style and formatting guide <https://www.nature.com/documents/commsj-phys-style-formatting-guide-accept.pdf> Communications Sustainability formatting guide .

*** DATA: Communications Sustainability endorses the principles of the Enabling FAIR data project (<http://www.copdess.org/enabling-fair-data-project/>). We ask authors to make the data that support their conclusions available in permanent, publicly accessible data repositories. (Please contact the editor if you are unable to make your data available).

All Communications Sustainability manuscripts must include a section titled "Data Availability" at the end of the Methods section or main text (if no Methods). More information on this policy, is available at <http://www.nature.com/authors/policies/data/data-availability-statements-data-citations.pdf>.

If a community resource is unavailable, data can be submitted to generalist repositories such as <https://figshare.com/> or <http://datadryad.org/> Dryad Digital Repository. Please provide a unique identifier for the data (for example a DOI or a permanent URL) in the data availability statement, if possible. If the repository does not provide identifiers, we encourage authors to supply the search terms that will return the data. For data that have been obtained from publicly available sources, please provide a URL and the specific data product name in the data availability statement. Data with a DOI should be further cited in the methods reference section.

REVIEWER COMMENTS:

Reviewer #1 (Remarks to the Author):

This is a very well-written, clear and relevant paper, I really enjoyed reading it. I do have a couple of points below that may be considered for improvement.

1) The methodology differentiates between three stages at which fish is "lost" in the supply chain: when it is not considered as food at all, a post-harvest loss and a loss of by-products. However, the three stages could be better defined and this introduced a concern that there is a risk of double counting.

A clear differentiation is made between post-harvest losses and by-products but commonly post harvest losses already contain some by-products. In other words, it is questionable if b (the consumed proportion of the food-use fish) consists of all whole fish, and if therefore c can be assumed.

2) There is no indication of discarded fish (at sea) and it is unclear how harvested fish is defined. Can you define if "harvested" fish is what is brought to land or what is captured at sea? In the latter case, post-harvest losses would include fish that is discarded at sea? And what about processing at-sea?

3) Reference no. 4 is used to calculate the baseline fractions of fish use which is the basis for the fractions available after improvements. However, ref. 4 calculates most values for aquatic foods, which includes e.g., crustaceans and molluscs. Is this considered in your calculations? Also, in the reference list, reference no. 4 is FAO but the only reference I could find was from World Economic Forum.

4) The additional available fish is expressed as "portions of fish available" but it is nowhere defined how large a portion is assumed to be. This can differentiate among consumers, so it should be specified.

5) The Discussion could benefit from a bit more comparison with the existing literature, i.e., as a means of validation of the results. It currently has some repetition of the results.

6) Do I understand correctly that only the protein isolate from by-products is considered to be re-used as human food? In my opinion this underestimates the potential of byproducts and this is at least a point to address in the discussion.

7) The methodology section stops abruptly and repeats part of the introduction; I think something went wrong in the formatting here.

8) Include a legend for the colors in Figure 2.

Reviewer #2 (Remarks to the Author):

This paper is to address ways to increase the amount of seafood available for consumers through the reduction of "waste". There are many papers that have referenced ways to accomplish this, and this paper builds on this body of literature. This is a needed topic of study given the lost and waste of all food types, not just seafood.

While this is a needed study, the authors consider and unusual view of waste and loss. They define PHL as "global fish catch is lost or wasted along supply chains". This is fine, although terrestrial agriculture defines loss and anything discarded up to the point of leaving the processor, and waste as anything discarded at retail or consumer level. Love et al 2015 (<https://doi.org/10.1016/j.gloenvcha.2015.08.013>) have a very nice figure of discards along the value chain. Importantly, they bring up the concept of edible vs inedible portions. For example, lobster yields only 35% meat, so if sold whole, the consumer is the progenitor of that waste, where as if they are processed, this loss appears elsewhere. But no where can this be upcycled into human food, unless it is extracted for stocks etc. But this paper also does not address crustaceans. So they need to consider the dressing percents for salmon (60%) vs tilapia (35%). This lack of definition of PHL is why in the abstract it states "Globally, only 54% of harvested fish is consumed directly by people", yet in the Main, it states "Approximately 35% of the global fish catch is lost or wasted" and in the results, "FAO's global food loss estimates for fisheries, which range from 25% to 35%". Numbers are floating around, and it is difficult to tract them all, and determine which are ranges, estimates, or calculated values.

Additionally, some of the "PHL" component's are actually food sources themselves. The first step in figure 1 is non food use (M2) that is called out as fish meal and fish oil. Fish oil can be used directly for pills, and thus is meeting a human nutritional requirement. Fish meal can be created into fish balls, but is more commonly incorporated into animal feeds that created additional nutrition. It may not be an efficient use of resources, but it is not be disposed of as the PHL moniker would indicate.

Finally, when calculating nutrition gained from up-cycling discards, it is not fair to assume herring as the metric for all products. As the authors point out " Compared to lean white fish such as cod, herring by-products offer a more diverse micronutrient profile, while also serving as a realistic substrate for use in staple or fortified food applications". What of all the white fish waste? This also does not account for the 15% (likely under-reported) of fresh water fish that are caught that have a much lower nutrient (considering omega 3 fatty acids) profile. The use of herring as the benchmark will over inflate this value.

The adoption of technology assumes that an increase in technology leading to increased food for consumers will equally reduce PHL at each node. The reduction fisheries are systematically entrenched, so will technology actually reduce that proportion? Why are these technologies applied evenly to all PHL across the board?

The case study and regional dynamics analysis sections are difficult to follow. Is it fair to lump Asia as a singular region? As for the data, there are no case study references there, and only percents are provided for each region: here they are pasted-

x f
Africa Asia Europe North America Oceania South America

0% 41% 55% 58% 57% 40% 45%

10% 44% 57% 60% 59% 43% 48%

30% 49% 61% 65% 63% 49% 52%

50% 55% 66% 69% 68% 55% 58%

80% 65% 73% 76% 74% 66% 66%

This is merely having a start value for each area, and then applying a technology improvement percent evenly across the board. Are all technologies likely to be adoption with the same percent?
Line 443 states "examples cited in" and an nothing more. An important piece of this sentence is missing.

The authors state "All citations of numeric data or specific claims in the text correspond to the original sources. Those references were maintained in References for transparency and to allow readers to consult the original studies or reports for more detail.", yet refer to a Flickr story without providing a link. If they are relying of this type of data, then it should be listed in a data source file. There is no way to figure out how their starting level of waste for Africa or Asia is calculated.

It is important to get more nutritious food to people. But this is a systemic problem, and pointing out where loss and waste occurs is important. Technology is an important tool, but given the abundance of technology currently available, we still have reduction fisheries, and a lack of a cold chain in Africa. Furthermore, people do not like to eat small oily fish. How do we create a wave of change for food preferences. This will be a big step to changing the way we eat seafood and as a result, our overall efficiency.

** Visit Nature Portfolio's author and reviewers' website at www.nature.com/authors for information about policies, services and author benefits**

Communications Sustainability is committed to improving transparency in authorship. As part of our efforts in this direction, we are now requesting that all authors identified as 'corresponding author' create and link their Open Researcher and Contributor Identifier (ORCID) with their account on the Manuscript Tracking System prior to acceptance. ORCID helps the scientific community achieve unambiguous attribution of all scholarly contributions. You can create and link your ORCID from the home page of the Manuscript Tracking System by clicking on 'Modify my Springer Nature account' and following the instructions in the link below. Please also inform all co-authors that they can add their ORCIDs to their accounts and that they must do so prior to acceptance.

Version 1:

Decision Letter:

Dear Dr Willer,

Your revised manuscript titled "Technological innovations can reduce post-harvest fish losses and sustainably improve nutritional and economic outcomes" has now been seen by 2 reviewers, whose comments are appended below. You will see that they recognise the improvements made. However, they have raised some concerns that must be addressed. In light of these comments, further revisions will be required before we can further consider the manuscript for publication. We would, however, be interested in considering a revised version that fully addresses these concerns.

In particular, we require that you:

- fully acknowledge the limitations of your approach,
- address methodological and data clarity as highlighted by the reviewers.

We hope you will find the reviewers' comments useful as you decide how to proceed. Should additional work allow you to address these criticisms, we would be happy to look at a substantially revised manuscript. If you choose to take up this option, please either highlight all changes in the manuscript text file, or provide a list of the changes to the manuscript with your responses to the reviewers.

When resubmitting, please provide a point-by-point response to the reviewers' comments. Please submit your responses as a separate file, distinct from your cover letter where you can add responses to the Editors' comments that you do not want to be made available to the reviewers. Word files are preferred. We recommend that any figures, tables or graphs that are included in the response to reviewers are also included in the main article or Supplementary Information.

If the revision process takes significantly longer than three months, we will be happy to reconsider your paper at a later date, as long as nothing similar has been accepted for publication at Communications Sustainability or published elsewhere in the meantime.

Please use the following link to submit your revised manuscript, point-by-point response to the reviewers' comments with a list of your changes to the manuscript text (which should be in a separate document to any cover letter), a tracked-changes version of the manuscript (as a PDF file) and any completed checklist:

Link Redacted

Please do not hesitate to contact us if you have any questions or would like to discuss the required revisions further. Thank you for the opportunity to review your work.

Best regards,

Alice Drinkwater, PhD
Consulting Editor
Communications Sustainability
Associate Editor
Communications Earth & Environment

EDITORIAL POLICIES AND FORMAT

If you decide to resubmit your paper, please ensure that your manuscript complies with our editorial policies and complete and upload the checklist below as a Related Manuscript file type with the revised article:

- Behavioural and social science
- Ecological, evolutionary & environmental sciences
- Life sciences

For your information, you can find some guidance regarding format requirements summarized on the following checklist: (<https://www.nature.com/documents/commsj-phys-style-formatting-checklist-article.pdf>) and formatting guide (<https://www.nature.com/documents/commsj-phys-style-formatting-guide-accept.pdf>).

REVIEWER COMMENTS:

Reviewer #1 (Remarks to the Author):

Thank you for carefully revising this manuscript. To my regard, the concerns were addressed accordingly. From the answer to my first point, and figure 1, I draw the conclusion that the study assumes that none of the PHL could re-enter the food chain, while the total volume of PHL could be decreased. I would argue that innovations could also target the revalorisation of PHL.

A couple small things to make the methods section slightly more clear:

Line 340: I would suggest to remove "edible" for the part "remains available as edible fish" because the following sentence suggests that it is actually a stream still containing edible and unedible and only after applying c, the distinction is made between the two.

Line 354: make the percentages fractions instead, because they should be used in the equation as fractions.

Line 357-359: rename alim etc. to amax, bmax, etc. I think it is more intuitive to use this indication as it is the maximum that could be reached if x was 1.

Reviewer #2 (Remarks to the Author):

The rebuttal to reviewers was interesting as many of the questions noted in the first submission, were explained as being best case scenarios. I noted the broad focus of technology adoption with a large similarity of value based on content, and nutritional benefits begin modeled on herring. Herring is by far not the most significant fish, but it does have a high nutritional value. The authors then states that these are best case scenarios. That is fine, but the manuscript remains focused on these best case scenarios (they added the statement of herring as an upperbound) but without qualifying the likelihood of this happening, or what a more honest value will be.

The other issue is that this analysis appears to be predicated on assuming a single use level without assessing the difference in country participation in global fisheries. the statements:

"The 74% potential represents a theoretical upper bound under a harmonised, globally comparable⁶⁵ adoption scenario."

"The values of a_0 , b_0 , and d_0 varies³⁶⁴ across continents, while globally, $a_0 = 89\%$, $b_0 = 81\%$, and $d_0 = 30\%$. The value of $c = 65\%$ is³⁶⁵ considered as a constant throughout the analysis.

This approach does not consider the weighting of fisheries and gives a greater weight to low values for Bangladesh, and "Southeast Asia, artisanal supply chains experience elevated losses".

But for all the discussion of understanding PHL at a global level, they address the point above by stating "In addition, treating technological adoption as proportional and harmonised across nodes is a deliberate²⁹⁸ simplification to obtain globally comparable bounds" - a statement I disagree with. I think the deliberate simplification will reduce future attention.

How will the cost of implementation of the technology change the price? Odd that for an economic analysis, cost externalities were not included.

It is hard to tell where the data were derived from (their data frame in their github is just the percent value they state in the paper). There are not the list of values and countries for which they collected values and then the average.

³⁶⁴ - odd phrasing - condition is calculated by Equation 3, where a_0 , b_0 , and d_0 are based literatures.

** Visit Nature Portfolio's author and reviewers' website at www.nature.com/authors for information about policies, services and author benefits**

Communications Sustainability is committed to improving transparency in authorship. As part of our efforts in this direction, we are now requesting that all authors identified as 'corresponding author' create and link their Open Researcher and Contributor Identifier (ORCID) with their account on the Manuscript Tracking System prior to acceptance. ORCID helps the scientific community achieve unambiguous attribution of all scholarly contributions. You can create and link your ORCID from the home page of the Manuscript Tracking System by clicking on 'Modify my Springer Nature account' and following the instructions in the link below. Please also inform all co-authors that they can add their ORCIDs to their accounts and that they must do so prior to acceptance.

If you experience problems in linking your ORCID, please contact the Platform Support Helpdesk.

Version 2:

Decision Letter:

Dear Dr Willer,

Your manuscript titled "Technological innovations can reduce post-harvest fish losses and sustainably improve nutritional and economic outcomes" has now been seen by our reviewers, whose comments appear below. In light of their advice we are delighted to say that we are happy, in principle, to publish a suitably revised version in Communications Sustainability.

We therefore invite you to revise your paper one last time to address the remaining concerns of our reviewers, which pertain to acknowledging limitations. At the same time we ask that you edit your manuscript to comply with our format requirements and to maximise the accessibility and therefore the impact of your work.

EDITORIAL REQUESTS:

****Please take care to match our formatting and policy requirements. We will check revised manuscript and return manuscripts that do not comply. Such requests will lead to delays. ****

SUBMISSION INFORMATION:

OPEN ACCESS:

Communications Sustainability is a fully open access journal. Articles are made freely accessible on publication. For further information about article processing charges, open access funding, and advice and support from Nature Portfolio, please visit <https://www.nature.com/commssustain/open-access>

Link Redacted

Best regards,

Alice Drinkwater, PhD
Consulting Editor
Communications Sustainability
Associate Editor
Communications Earth & Environment

REVIEWERS' COMMENTS:

Reviewer #2 (Remarks to the Author):

The sentence added in the results - "It should be noted that rather than predicting realised outcomes, the analysis is designed to establish upper⁵² bounds on what could be achieved under idealised and harmonised adoption conditions, thereby bounding the scale⁵³ of opportunity embedded in post-harvest optimisation." needs to go in the abstract. The abstract still reads as though this is achievable, and not an upper bound that does not assess true cost of technology implementation. Please read what you wrote in the paragraph at line 305 and have the abstract be consistent with this - especially the statement "interpret the results as upper limits on recoverable potential rather than expected outcomes"

The sentence "It should¹⁷⁰ be noted that the price effects reported here reflect efficiency gains conditional on successful technology adoption and¹⁷¹ do not incorporate capital, operating, or transition costs associated with implementation" needs to be followed by something along the lines of - - This analysis also does not account for the system wide improvements at a region and country level that would need to occur in order for the electrical grid to be stable enough for improvements in energy intensive technology improvements such as cold chain storage.

The sentence at line 305 is an over promise "While this analysis provides robust, globally scalable estimates of system-level potential," - This analysis provides a best case scenario of potential improvement in absence of realized costs and externalities.

** Visit Nature Portfolio's author and reviewers' website at www.nature.com/authors for information about policies, services and author benefits**

POINT-BY-POINT RESPONSE TO REVIEWERS' COMMENTS

Reviewer #1:

This is a very well-written, clear and relevant paper, I really enjoyed reading it. I do have a couple of points below that may be considered for improvement.

1) The methodology differentiates between three stages at which fish is “lost” in the supply chain: when it is not considered as food at all, a post-harvest loss and a loss of by-products. However, the three stages could be better defined and this introduced a concern that there is a risk of double counting.

A clear differentiation is made between post-harvest losses and by-products but commonly post harvest losses already contain some by-products. In other words, it is questionable if b (the consumed proportion of the food-use fish) consists of all whole fish, and if therefore c can be assumed.

Author response: Thank you very much for this thoughtful comment. We confirm that no double-counting occurs, while we agree that the distinctions among the three categories of non-food use, post-harvest loss, and by-products must be defined clearly to avoid any perceived overlap or double-counting.

We would like to clarify that, in our calculations, we defined the three stages accords to FAO’s SOFIA 2024 utilisation framework and WEF white paper, and kept the accounting mutually exclusive.

First, “non-food uses” (parameter $1 - a$) refer to whole fish that never enter the food chain (e.g., fish reduced to fishmeal/oil, bait, ornamental uses, etc.). These volumes are, by definition, outside the food pathway and therefore cannot generate edible by-products for human consumption in our accounting. FAO and the WEF reports that approximately 89% of fishes (demersal fishes, small pelagics, tunas, salmons, and other fishes) are directed to food and about 11% to non-food uses, most of which are reduced to fishmeal and fish oil; this is the boundary we adopt.

Second, our definition of “post-harvest loss” (parameter $1 - b$) follows the WEF white paper methodology, in which losses during capture, landing, and processing include both edible and inedible fractions that fail to reach consumers. This represents the complete biomass reduction along the food-use pathway, i.e., all mass no longer available for human consumption, and therefore intentionally uses a broader accounting boundary than the FAO definition of food loss, which focuses only on avoidable edible loss.

Third, by-products (parameter $1 - c$) are then separated from this post-harvest loss term as the structural inedible fractions of fish that remain after processing (heads, bones, viscera, skin). Their potential revalorisation into food is accounted for separately via parameter d . In other words, inedible mass is first counted once within post-harvest loss and then explicitly partitioned into by-products, ensuring mutually exclusive material flows.

To address the specific doubt, whether b (the consumed proportion of food-use fish) consists of all whole fish, and whether c can therefore be assumed, we clarify that b is expressed in whole-fish equivalents, purely as a bookkeeping device. Conceptually, b removes not only the edible portion that is lost or spoiled from that stream, but also trimmings and non-edible fractions. Following this, c accounts for the inedible fractions of the whole fish not lost between landing and the consumer. This separation prevents any overlap.

The net utilisation equation we use, $f = a \cdot b \cdot (c + (1 - c) \cdot d)$, therefore maps directly onto FAO’s hierarchy (food vs. non-food, edible vs. by-product, by-product reuse) without double counting.

In the revised manuscript, we added a short paragraph in the Methods section explicitly stating these mutually exclusive flows for each boundary: (i) partitioning between food and non-food uses; (ii) losses along the food-use pathway; and (iii) generation and reuse of by-products during processing.

2) There is no indication of discarded fish (at sea) and it is unclear how harvested fish is defined. Can you define if “harvested” fish is what is brought to land or what is captured at sea? In the latter case, post-harvest losses would include fish that is discarded at sea? And what about processing at-sea?

Author response: Thank you for your comment. In the revised Method section, we clarify that, consistent with FAO SOFIA 2024, “harvested fish” refers to landed production (i.e., catch that is brought to shore and enters FAO’s production/utilisation statistics). Fish that are caught but discarded at sea are therefore not part of the harvested volume in our model, do not enter FAO utilisation accounts, and lie outside our system boundary.

3) Reference no. 4 is used to calculate the baseline fractions of fish use which is the basis for the fractions available after improvements. However, ref. 4 calculates most values for aquatic foods, which includes e.g., crustaceans and molluscs. Is this considered in your calculations? Also, in the reference list, reference no. 4 is FAO but the only reference I could find was from World Economic Forum.

Author response: Thank you for raising this clarification request. Our baseline fractions of fish use are indeed derived from Reference 4. In that reference, aquatic foods are categorised into eight groups: demersal fishes, small pelagics, tunas, shrimp and prawns, molluscs, crustaceans, salmons, and other fishes. Because our study focuses specifically on fish, we excluded shrimp and prawns, molluscs, and crustaceans from the calculations. Only the fish-relevant categories were used when establishing the baseline fractions. This ensures that the fractions used in our model reflect fish utilisation patterns rather than the broader category of aquatic foods.

We also acknowledge the reviewer’s note regarding the discrepancy in the citation. The originally listed Reference 4 was incorrectly attributed to FAO, while the data source we used was from the World Economic Forum. We have corrected this in the revised manuscript to accurately reflect the source.

4) The additional available fish is expressed as “portions of fish available” but it is nowhere defined how large a portion is assumed to be. This can differentiate among consumers, so it should be specified.

Author response: Thank you for highlighting the need to clearly define what constitutes a “portion of fish.” Although we noted in the *Nutritional and public health benefits* section that “If recovered, these losses could yield approximately 850 million additional 100 g portions per day...,” we agree that this definition should be stated more explicitly. In the revised manuscript, we have added a sentence in the Methods section specifying that a portion is defined as 100 grams of edible fish and clarifying that all results are expressed in 100-gram edible portions unless otherwise stated.

5) The Discussion could benefit from a bit more comparison with the existing literature, i.e., as a means of validation of the results. It currently has some repetition of the results.

Author response: Thank you for the suggestion to strengthen the Discussion through comparison with existing literature. In response, we have added a new paragraph summarizing

recent field-based and review studies on post-harvest fish losses and intervention outcomes. These references provide empirical loss ranges and documented reduction effects, which align closely with the magnitude and sensitivity of losses represented in our model. This added context helps validate the realism and relevance of our findings against observed data.

6) Do I understand correctly that only the protein isolate from by-products is considered to be re-used as human food? In my opinion this underestimates the potential of byproducts and this is at least a point to address in the discussion.

Author response: Thank you for pointing this out. We have clarified in the revised manuscript that protein isolate is used only as an example of a food-upcycling pathway rather than as the exclusive or dominant route for valorising by-products. The intention was to provide a single, standardised metric that could be modelled globally, not to imply that other edible fractions or food applications are excluded.

To avoid underestimating the broader potential of by-products, we now explicitly note in the Discussion that fish by-products contain multiple components suitable for human consumption, such as oils, collagen, gelatin, mechanically recoverable meat, and mineral-rich fractions, and that these can enter a variety of food products depending on regional practices and technological capacity. Because our analysis focused on one illustrative pathway to maintain comparability across regions, we acknowledge that the results represent a conservative estimate. We have added a brief explanation in Discussion highlighting that the actual food-use potential of by-products is likely higher when considering the wider set of available applications.

7) The methodology section stops abruptly and repeats part of the introduction; I think something went wrong in the formatting here.

Author response: We appreciate the reviewer for pointing this out. The abrupt ending and repeated text in the Methodology section were caused by a formatting error during document compilation. We have carefully revised it.

8) Include a legend for the colors in Figure 2.

Author response: Thank you for pointing this out. We have now added a color legend to Figure 2.

Reviewer #2:

This paper is to address ways to increase the amount of seafood available for consumers through the reduction of "waste". There are many papers that have referenced ways to accomplish this, and this paper builds on this body of literature. This is a needed topic of study given the lost and waste of all food types, not just seafood.

While this is a needed study, the authors consider an unusual view of waste and loss. They define PHL as "global fish catch is lost or wasted along supply chains". This is fine, although terrestrial agriculture defines loss and anything discarded up to the point of leaving the processor, and waste as anything discarded at retail or consumer level. Love et al 2015 (<https://doi.org/10.1016/j.gloenvcha.2015.08.013>) have a very nice figure of discards along the value chain. Importantly, they bring up the concept of edible vs inedible portions. For example, lobster yields only 35% meat, so if sold whole, the consumer is the progenitor of that waste, where as if they are processed, this loss appears elsewhere. But no where can this be upcycled into human food, unless it is extracted for stocks etc. But this paper also does not address crustaceans. So they need to consider the dressing percents for salmon (60%) vs tilapia (35%). This lack of definition of PHL is why in the abstract it states "Globally, only 54% of harvested fish is consumed directly by people", yet in the Main, it states "Approximately 35% of the global fish catch is lost or wasted" and in the results, "FAO's global food loss estimates for fisheries, which range from 25% to 35%". Numbers are floating around, and it is difficult to tract them all, and determine which are ranges, estimates, or calculated values.

Author response: Thank you for this careful and constructive comment. We agree that differences in terminology surrounding loss, waste, edible versus inedible fractions, and species-specific yields have led to confusion in the literature. To address this, we explicitly align our accounting boundaries with the WEF Aquatic FLW Annex (2024), while maintaining consistency with FAO's SOFIA 2024 utilisation framework. In the revised Methods, we clearly define each term and trace the fate of biomass through mutually exclusive flows.

In our study, post-harvest loss (PHL) is defined broadly as the reduction of biomass within the food-use pathway, including both edible and structural inedible fractions of fish that fail to reach consumers during capture, landing, transport, storage, and processing. This aligns with the WEF framework, in which loss during production and processing includes edible and inedible portions, because both reduce aquatic food availability for human consumption. This broader boundary differs from FAO's narrower definition of "food loss" (avoidable edible loss only), and we now make this distinction explicit in the manuscript.

Structural inedible mass (e.g., bones, viscera, skin) is first included once in PHL, and is then explicitly partitioned into by-products during processing. These by-products may subsequently be valorised for food (e.g., minces, collagen/gelatin ingredients, fish oil for direct consumption, fermented or traditional regional foods) or diverted to non-food uses. This prevents double counting while preserving the ability to quantify food recovered through by-product utilisation.

We further clarify how species-specific edible yields interface with PHL and how our modelling scope relates to the species groups reported in the WEF Aquatic FLW Annex. The WEF framework distinguishes several major groups – demersal fishes, small pelagics, tunas, salmon, "other fishes", crustaceans and molluscs. In this paper, our explicit focus is on finfish, and we therefore use only the finfish-related categories from the WEF dataset (demersal fishes, small pelagics, tunas, salmon and other fishes) when parameterising the model. As the reviewer notes, species such as lobsters have low edible yields (~35%), whereas salmon or tilapia have substantially higher yields. In our framework, such yield differences are captured through the

edible-portion parameter, rather than being treated as loss, because yield is a biological and processing property, not an avoidable post-harvest loss metric. Crustaceans and molluscs are therefore outside the scope of this finfish-focused analysis, rather than being omitted from the accounting, and we now state this explicitly in the Methods to avoid any ambiguity.

This clarification also resolves the reviewer’s concern about the multiple global figures cited (e.g., “54% of harvested fish is consumed,” “35% is lost or wasted”). These are now consistently presented as:

- (1) the share of the catch that enters food (after non-food uses are removed),
- (2) the share of biomass lost within that pathway, and
- (3) the share ultimately consumed.

Together, these revisions ensure that all numbers reported are traceable, non-overlapping, and directly comparable to FAO and WEF frameworks.

Additionally, some of the "PHL" component's are actually food sources themselves. The first step in figure 1 is non food use (M2) that is called out as fish meal and fish oil. Fish oil can be used directly for pills, and thus is meeting a human nutritional requirement. Fish meal can be created into fish balls, but is more commonly incorporated into animal feeds that created additional nutrition. It may not be an efficient use of resources, but it is not be disposed of as the PHL moniker would indicate.

Author response: Thank you for your comment. We agree that several streams classified as “non-food use” in FAO’s utilisation framework, in particular fish oil and, in some cases, fishmeal, can ultimately contribute to human nutrition, either directly via supplements or indirectly through livestock and aquaculture feeds. Our use of the term “non-food use” therefore follows FAO’s utilisation categories, which are based on whether products enter markets as edible fish, rather than whether they ultimately support human nutrition through other pathways.

To avoid confusion with post-harvest loss, we clarify that non-food utilisation is not counted as PHL in our model. PHL refers to the reduction of biomass that would otherwise remain available as edible fish products, including both edible and structural inedible fractions that fail to reach consumers and are not redirected into the by-product valorisation pathway. In contrast, reduction into fishmeal and fish oil remains within the FAO-defined non-food utilisation branch and is not interpreted as waste or loss in our accounting.

This separation ensures that PHL is not conflated with productive non-food uses, while preserving the broader system perspective recommended by the WEF FLW Annex — where loss at production and processing stages is tracked as a reduction of biomass available for edible fish, even if some nutritional value is later recovered via other routes. We now emphasise this distinction in the Methods and figure captions to avoid implying that non-food utilisation equates to waste.

Finally, when calculating nutrition gained from up-cycling discards, it is not fair to assume herring as the metric for all products. As the authors point out " Compared to lean white fish such as cod, herring by-products offer a more diverse micronutrient profile, while also serving as a realistic substrate for use in staple or fortified food applications". What of all the white fish waste? This also does not account for the 15% (likely under-reported) of fresh water fish that are caught that have a much lower nutrient (considering omega 3 fatty acids) profile. The use of herring as the benchmark will over inflate this value.

Author response: Thank you for this important clarification on nutrient modelling. We agree that using herring as a single benchmark species can bias results upward because herring has a comparatively rich fatty-acid and micronutrient profile that is not representative of many lean whitefish or a large share of freshwater species. In the revised Method section, we clarify that herring was used as an illustrative case due to relatively strong by-product composition data and its practical relevance in existing fortification and processing applications, not as a universal proxy for all taxa or regions. This distinction is now explicit so that readers do not interpret the herring case as a system-wide average. Additionally, to align with FAO’s utilisation framework and species breadth, we emphasise in the revised Method section that nutrient recovery from up-cycling is intrinsically species- and product-form-dependent. FAO documents wide heterogeneity in production across species groups and environments, with substantial volumes from inland systems and diverse taxa entering food channels; these differences imply materially different edible yields and nutrient profiles compared with oily pelagic fish such as herring. Consequently, where species mixes are dominated by lean whitefish or freshwater taxa, the achievable EPA/DHA and fat-soluble vitamin recovery from by-products will generally be lower than the herring-based illustration. We have added language in the revised Method section stating that our herring example represents an upper-bound, “species-richness ceiling” for certain nutrients rather than a global mean, and that realised gains depend on regional species composition and processing characteristics.

Furthermore, we now note in the Discussion section that future work should incorporate species-weighted nutrient modelling using region-specific species baskets and dressing yields to produce more granular estimates.

The adoption of technology assumes that an increase in technology leading to increased food for consumers will equally reduce PHL at each node. The reduction fisheries are systematically entrenched, so will technology actually reduce that proportion? Why are these technologies applied evenly to all PHL across the board?

Author response: Thank you for raising this important point about how technology adoption is handled in our model. You are correct that treating technological improvements as reducing PHL uniformly across all nodes of the value chain is a simplifying assumption, and we agree that real-world reductions would vary substantially depending on species, region, supply-chain structure, and the degree to which reduction-oriented fisheries are institutionally entrenched.

In the revised manuscript, we now clarify that our modelling framework is designed to estimate theoretical maximum recoverable potential under a harmonised, globally comparable scenario. This requires standardising the effect of technological adoption across nodes so that regions with different supply-chain geometries can be compared in a single model. However, we fully acknowledge that in practice technologies do not reduce losses uniformly. Cold chain improvements disproportionately affect early-stage microbial spoilage; improved handling reduces mechanical damage at landing sites; smoking/drying technologies mainly reduce losses during processing; and valorisation technologies affect later-stage by-product utilisation rather than early-stage losses. We now explicitly state that our assumption of evenly applied reduction coefficients is a simplification that helps isolate the cumulative system-level effect of technology portfolios rather than reflect node-specific real-world reductions.

Regarding reduction fisheries: you are absolutely right that these supply chains are structurally locked in and unlikely to be influenced by post-harvest technologies in the same way as fish destined for human consumption. We therefore now add language explaining that our model

applies technology adoption only to the portion of the catch intended for human consumption, and that entrenched reduction fisheries represent a structural constraint where technological interventions may have limited influence. This is an important distinction, and we now emphasise that the proportion of biomass going into reduction pathways is largely a consequence of market demand and industry configuration, not of post-harvest inefficiency. As such, our model does not implicitly assume that technologies would reallocate fish away from reduction sectors.

Together, these revisions clarify for readers that the model is meant to estimate systemic potential under harmonised adoption scenarios, not to predict highly granular real-world reduction patterns at each node. We also state in the Discussion that more detailed, species- and node-specific modelling will be needed in future work to capture heterogeneous impacts across the value chain.

The case study and regional dynamics analysis sections are difficult to follow. Is it fair to lump Asia as a singular region? As for the data, there are no case study references there, and only percents are provided for each region: here they are pasted-

	Africa	Asia	Europe	North America	Oceania	South America
0%	41%	55%	58%	57%	40%	45%
10%	44%	57%	60%	59%	43%	48%
30%	49%	61%	65%	63%	49%	52%
50%	55%	66%	69%	68%	55%	58%
80%	65%	73%	76%	74%	66%	66%

This is merely having a start value for each area, and then applying a technology improvement percent evenly across the board. Are all technologies likely to be adoption with the same percent?

Author response: Thank you for raising these important concerns about the case-study framing, regional aggregation, and modelling of technology adoption. We recognise that the way regional dynamics are currently presented can give the impression of oversimplification, and we agree that several clarifications are needed.

First, we agree that grouping all of Asia into a single region is a coarse simplification. Asia is the most diverse aquatic food region globally, spanning highly industrialised value chains (e.g., East Asia), small-scale artisanal systems (e.g., South and Southeast Asia), and mixed capture–aquaculture systems with widely varying loss mechanisms. In the revised manuscript, we now explain that Asia is used as an aggregated region only because of the structure of the FAO loss dataset, which reports post-harvest loss at continental or subcontinental levels but not at country scale. We also note that the six-region division we use (Africa, Asia, Europe, North America, Oceania, South America) is consistent with the regional segmentation used by FAO and the World Economic Forum in their respective global fisheries and food-system analyses. We have added text emphasising Asia’s internal heterogeneity and that regional mean values should not be interpreted as uniform conditions across the continent. We also highlight that future work should disaggregate Asia where country-level or supply-chain-level PHL data become available.

Second, we now clarify the purpose of the “case study” section. This section is meant to demonstrate how the model behaves when applied to a specific region, not to provide empirical validation for any single country or supply chain. We have added text to the Methods and Discussion explaining that the case study uses FAO-reported regional averages as starting conditions and that these numbers are not intended as primary data for any particular fishery. We have also clarified in the Results that the current analysis is not meant to represent real-world case-study documentation, but rather a scenario-based exploration of potential improvements under uniform parameterisation.

Finally, regarding technology adoption, you are correct that applying a single adoption percentage across an entire region and across all technologies is a simplification. We have now explicitly added text stating that this is a scenario design choice, intended to isolate the system-wide sensitivity of PHL to adoption intensity, rather than a prediction that all technologies will diffuse at the same rate. In the revised manuscript, we explain that cold chain upgrades, handling improvements, drying innovations, and valorisation technologies would not be adopted uniformly across geographies or value-chain nodes. We now emphasise that our approach is designed to generate a harmonised, comparable scenario rather than model the diffusion of specific technologies in specific countries.

Together, these amendments make it clear that (1) the aggregated regions reflect FAO data constraints, (2) the “case study” is a scenario illustration rather than field evidence, and (3) uniform adoption rates are a modelling convenience rather than a real-world assumption.

Line 443 states "examples cited in" and an nothing more. An important piece of this sentence is missing.

Author response: We appreciate the reviewer for pointing this out. It was a formatting error regarding field function of Microsoft Word during document compilation. We have carefully revised it.

The authors state "All citations of numeric data or specific claims in the text correspond to the original sources. Those references were maintained in References for transparency and to allow readers to consult the original studies or reports for more detail.", yet refer to a Flickr story without providing a link. If they are relying of this type of data, then it should be listed in a data source file.

Author response: Thank you for this helpful comment. We appreciate the reviewer’s attention to source quality and transparency. In the revised manuscript, we have replaced the previously cited anecdotal materials (e.g., the FAO Flickr story) with peer-reviewed and technical sources that provide quantitative evidence of post-harvest loss interventions, particularly within Sub-Saharan African fisheries.

We would also like to note that these examples illustrate only a small subset of the literature we consulted; they are now presented solely to demonstrate the types of empirically based interventions that informed our scenario analyses. All numeric data and specific claims retained in the text are now fully traceable to properly archived sources cited in the reference list.

There is no way to figure out how their starting level of waste for Africa or Asia is calculated.

Author response: The starting level is calculated according to Ref. 4, i.e., the white paper report provided by World Economic Forum, where the data of all continents are respectively presented.

It is important to get more nutritious food to people. But this is a systemic problem, and pointing out where loss and waste occurs is important. Technology is an important tool, but given the abundance of technology currently available, we still have reduction fisheries, and a lack of a cold chain in Africa. Furthermore, people do not like to each small oily fish. How do we create a wave of change for food preferences. This will be a big step to changing the way we eat seafood and as a result, our overall efficiency.

Author response: Thank you for raising this important broader point. We agree that increasing the nutritional contribution of aquatic foods is ultimately a systemic challenge, and not something that technological improvements alone can resolve. Your point regarding consumer preferences for small oily fish is especially important. Despite their high nutrient density, species such as sardine, anchovy, and herring are undervalued or less preferred in many markets. In the revised Discussion section, we note that improvements in post-harvest efficiency must be complemented by efforts that increase the desirability and acceptability of nutrient-rich species, for example through product reformulation and initiatives that familiarise consumers with these foods.

POINT-BY-POINT RESPONSE TO REVIEWERS' COMMENTS

Reviewer #1:

Thank you for carefully revising this manuscript. To my regard, the concerns were addressed accordingly. From the answer to my first point, and figure 1, I draw the conclusion that the study assumes that none of the PHL could re-enter the food chain, while the total volume of PHL could be decreased. I would argue that innovations could also target the revalorisation of PHL.

We thank the reviewer for this helpful interpretation. We agree that some innovations can target the recovery/revalorisation of streams that would otherwise be counted as post-harvest loss (e.g., secondary processing of downgraded product). In our current framework, re-entry into the human-food system is represented only through by-product utilisation (parameter d), while biomass counted in post-harvest loss ($100\% - b$) is treated as unrecoverable within the model boundary.

To avoid any ambiguity, we have now explicitly stated this assumption in the Methods and noted in the Discussion that technologies enabling recovery from loss streams could further increase human-food availability beyond our estimates and would require an additional recovery term in future modelling. We further clarify that, while recovery from loss streams is conceptually possible, its inclusion at a global scale would require species- and context-specific data on food-safety-constrained recovery yields that are currently unavailable; excluding this pathway therefore preserves model transparency and avoids boundary ambiguity with by-product valorization.”

A couple small things to make the methods section slightly more clear:

Line 340: I would suggest to remove "edible" for the part "remains available as edible fish" because the following sentence suggests that it is actually a stream still containing edible and unedible and only after applying c , the distinction is made between the two.

We agree and have removed “edible” here to avoid implying that the stream has already been partitioned into edible versus inedible fractions prior to applying c . The sentence now refers to “fish products” (prior to the edible-portion yield step).

Line 354: make the percentages fractions instead, because they should be used in the equation as fractions.

Thank you for this good point for clarity and consistency. In order to unify the expressions in Methods and Discussions, we have revised Equations 1–6 in the reversed direction, replacing fractions to percentages (e.g., $100\% - b$ instead of $1 - b$), as well as contents throughout Methods where parameters are used in equations.

Line 357-359: rename a_{lim} etc. to a_{max} , b_{max} , etc. I think it is more intuitive to use this indication as it is the maximum that could be reached if x was 1.

We agree that “ a_{lim} , b_{lim} , and d_{lim} ” is not intuitive enough, as the definition of a_{lim} and b_{lim} is different from d_{lim} . We have changed a_{lim} and b_{lim} to lim_a and lim_b , respectively, and changed d_{lim} to d_{max} as the reviewer suggested. We have renamed these parameters accordingly in the text and Equations 4–6, and updated the corresponding descriptions and values.

Reviewer #2:

The rebuttal to reviewers was interesting as many of the questions noted in the first submission, were explained as being best case scenarios. I noted the broad focus of technology adoption with a large similarity of value based on content, and nutritional benefits begin modeled on herring. Herring is by far not the most significant fish, but it does have a high nutritional value. The authors then states that these are best case scenarios. That is fine, but the manuscript remains focused on these best case scenarios (they added the statement of herring as an upperbound) but without qualifying the likelihood of this happening, or what a more honest value will be.

We agree with the reviewer that the manuscript focuses on best-case, system-level outcomes. In the revised version, we now explicitly frame all quantitative results as upper-bound estimates rather than predictions, and clarify that no claim is made regarding the likelihood of achieving these outcomes. We also make explicit that the use of Atlantic herring as the nutritional benchmark represents a nutrient-rich upper envelope rather than a globally representative species mix, and therefore intentionally biases nutritional estimates upward.

Estimating a single “more honest” realised value at the global scale would require strong assumptions about technology uptake probabilities across countries, species compositions, governance contexts, and supply-chain structures—parameters that are neither harmonised nor stable, and that vary substantially even within individual regions. Any global point estimate would therefore risk conveying a false sense of precision. We deliberately restrict the analysis to bounding system-level potential, and now discuss this choice and its implications more explicitly in Discussion, interpreting our findings as upper limits on recoverable potential rather than expected outcomes.

The other issue is that this analysis appears to be predicated on assuming a single use level without assessing the difference in country participation in global fisheries. the statements:

"The 74% potential represents a theoretical upper bound under a harmonised, globally comparable⁶⁵ adoption scenario."

"The values of a_0 , b_0 , and d_0 varies³⁶⁴ across continents, while globally, $a_0 = 89\%$ ^{1,4}, $b_0 = 81\%$ ⁴, and $d_0 = 30\%$ ⁴. The value of $c = 65\%$ ⁷⁵ is³⁶⁵ considered as a constant throughout the analysis.

This approach does not consider the weighting of fisheries and gives a greater weight to low values for Bangladesh, and "Southeast Asia, artisanal supply chains experience elevated losses".

We thank the reviewer for raising the issue of weighting and parameter aggregation. We would like to clarify that, in the model, baseline parameters describing allocation and utilisation efficiency (a_0 , b_0 , and d_0) are not treated as globally uniform. These parameters are explicitly specified at the continental level to reflect systematic regional differences in food-use allocation, post-harvest loss, and by-product utilisation. The corresponding values and literature sources are now transparently reported in Supplementary Table 1.

By contrast, the edible-portion yield (c) is treated as a global constant. This parameter represents a biological and processing yield rather than a supply-chain efficiency metric, and is therefore driven primarily by species morphology and processing conventions rather than country participation in global fisheries. At a global scale, c varies within a relatively narrow range across commercial finfish, making a species- and processing-averaged constant a reasonable approximation for system-level analysis.

We agree that we do not model differential adoption likelihood across individual countries or fisheries (e.g., Bangladesh vs. other producers). The baseline regional and global parameters are

taken directly from FAO and WEF aggregated utilisation accounts (Refs. 1 and 4), i.e., reported as shares at continental/global scales, rather than computed by the authors from country-level averages. Our harmonised adoption scenarios therefore bound system-level potential but do not represent production- or governance-weighted implementation pathways. We now clarify this modelling choice and its implications more explicitly in the Methods, Results, and Discussion.

But for all the discussion of understanding PHL at a global level, they address the point above by stating "In addition, treating technological adoption as proportional and harmonised across nodes is a deliberate²⁹⁸ simplification to obtain globally comparable bounds" - a statement I disagree with. I think the deliberate simplification will reduce future attention.

We thank the reviewer for this important point. We agree that describing harmonised adoption as a "deliberate simplification" without explicitly acknowledging its trade-offs may give the impression that this assumption is unproblematic or broadly applicable. In the revised manuscript, we have revised this wording to clarify that proportional and harmonised adoption is a modelling choice made to facilitate system-level comparability, but that it necessarily abstracts away from node-specific, country-specific, and context-dependent adoption dynamics.

We now emphasise that this assumption should not be interpreted as representative of real-world adoption pathways, and that its primary role is to bound system-level potential rather than to describe plausible implementation trajectories. We believe this clarification better reflects the limitations and appropriate use of the model.

How will the cost of implementation of the technology change the price? Odd that for an economic analysis, cost externalities were not included.

We thank the reviewer for raising this point. We agree that the economic analysis does not include the costs associated with implementing post-harvest technologies. The analysis is intended to capture efficiency gains arising from improved utilisation conditional on adoption, rather than to provide a full techno-economic assessment.

In the revised manuscript, we now explicitly clarify that the estimated price effects do not account for capital expenditures, operating costs, or transition costs associated with technology deployment, and should therefore not be interpreted as net welfare outcomes. We also note in the Discussion that upfront investment and financing constraints may delay or partially offset short-term price effects, particularly in small-scale and resource-constrained contexts.

It is hard to tell where the data were derived from (their data frame in their github is just the percent value they state in the paper). There are not the list of values and countries for which they collected values and then the average.

We thank the reviewer for raising this point. The continental and global values used in the analysis are not derived from country-level data aggregation performed by the authors. Instead, all baseline values are taken directly from FAO (Ref. 1) and the World Economic Forum Aquatic FLW Annex (Ref. 4), where utilisation and post-harvest loss metrics are already reported at global and continental scales.

Consequently, no list of country-level values or author-calculated averages exists for these parameters. Continental groupings follow the regional definitions adopted in the FAO and WEF frameworks (as illustrated in Fig. 3), including their treatment of transcontinental countries (e.g., Russia, Türkiye, Kazakhstan, Egypt, and French Guiana). To improve transparency, we now explicitly clarify the data provenance in the Methods and provide a supplementary table summarising the continental parameter values.

364 - odd phrasing - condition is calculated by Equation 3, where a_0 , b_0 , and d_0 are based literatures.

The sentence was begun with an equation, which may cause your misunderstanding. We have made the revision.